# Tunable Tamm plasmon cavity as a scalable biosensing platform for surface enhanced resonance Raman spectroscopy

Kandammathe Valiyaveedu Sreekanth [1] ✉, Jayakumar Perumal[1,4], U. S. Dinish[1,4], Patinharekandy Prabhathan[2,3], Yuanda Liu[1], Ranjan Singh [2,3] ✉, Malini Olivo[1,4] ✉ & Jinghua Teng [1] ✉

Surface enhanced Resonance Raman spectroscopy (SERRS) is a powerful technique for enhancing Raman spectra by matching the laser excitation wavelength with the plasmonic resonance and the absorption peak of bio-molecules. Here, we propose a tunable Tamm plasmon polariton (TPP) cavity based on a metal on distributed Bragg reflector (DBR) as a scalable sensing platform for SERRS. We develop a gold film-coated ultralow-loss phase change material ($Sb_2S_3$) based DBR, which exhibits continuously tunable TPP resonances in the optical wavelengths. We demonstrate SERRS by matching the TPP resonance with the absorption peak of the chromophore molecule at 785 nm wavelength. We use this platform to detect cardiac Troponin I protein (cTnI), a biomarker for early diagnosis of cardiovascular disease, achieving a detection limit of 380 fM. This scalable substrate shows great promise as a next-generation tunable biosensing platform for detecting disease biomarkers in body fluids for routine real-time clinical diagnosis.

Surface-enhanced Raman spectroscopy (SERS) is a promising analytical technique to investigate the molecular bonds of biomolecules by means of their unique vibrational fingerprints[1–3]. The Raman scattering cross-section of the molecules can be enhanced using metal nanoparticles (NP) supporting localized surface plasmon resonance (LSPR). In particular, the electromagnetic (EM) field enhancement occurs at the metal NP surface due to LSPR excitation and when the molecules adhere to the NP surface, they experience this enhanced EM near-field and scatter the light more effectively[1]. Using SERS, the enhancement factor at the order of $10^6$-fold and above has been realized using various plasmonic nanostructures[1–7]. SERS enhancement can be further improved by using shorter wavelength excitation sources since the Raman enhancement factor is directly proportional to the fourth power of the excitation frequency[8–10]. For this reason, SERS studies have been conducted even

using deep ultraviolet (DUV) laser sources[11,12]. However, photobleaching of biomolecules is a major issue with higher frequency excitation. Moreover, higher-frequency excitation and detection schemes are relatively complex and expensive. An alternate scheme for improving the enhancement factor is by combining SERS with the resonant Raman effect, which is known as Surface enhanced Resonance Raman spectroscopy[13–15] (SERRS). Using this method, an enhancement factor of $10^8$-fold and above can be achieved by exciting at (i) the plasmonic resonance wavelength, (ii) the absorption maximum wavelength of biomolecules, and (iii) the matching wavelength of plasmonic resonance and absorption maximum of biomolecules[13–19]. Among these, enhanced sensitivity with single molecule level detection has been realized by selecting the excitation frequency as the matching wavelength of plasmonic resonance and the absorption maximum of

[1]Institute of Materials Research and Engineering (IMRE), Agency for Science, Technology and Research (A*STAR), 2 Fusionopolis Way, Innovis #08-03, Singapore 138634, Republic of Singapore. [2]Division of Physics and Applied Physics, School of Physical and Mathematical Sciences, Nanyang Technological University, 21 Nanyang Link, Singapore 637371, Republic of Singapore. [3]Centre for Disruptive Photonic Technologies, The Photonic Institute, 50 Nanyang Avenue, Singapore 639798, Republic of Singapore. [4]Present address: A*STAR Skin Research Labs (A*SRL), Agency for Science, Technology and Research (A*STAR), 31 Biopolis Way, Nanos #07-01, Singapore 138669, Republic of Singapore. ✉e-mail: sreekanth@imre.a-star.edu.sg; ranjans@ntu.edu.sg; malini_olivo@asrl.a-star.edu.sg; jh-teng@imre.a-star.edu.sg

biomolecules[17–19]. Higher SERS enhancement factors have also been demonstrated by combining SERRS with shorter wavelength excitation sources[16–18]. Thus far, all reported plasmonic structures are passive, meaning that their plasmonic resonance is fixed for a specific excitation frequency. As a result, only a limited number of biomolecules whose absorption peak wavelength corresponds with the plasmonic resonance wavelength can be detected using SERRS. To address this issue, an active plasmonic substrate is necessary for SERRS to continuously tune its plasmonic resonance wavelength. This allows for the exact matching of the absorption peak wavelength of target biomolecules, achieving the highest possible enhancement. Additionally, a single SERRS substrate with multiple plasmonic modes can detect various biomolecules due to the tunability of the plasmonic resonance wavelength.

The development of ultrasensitive SER(R)S substrate highly depends on both top-down and bottom-up nanofabrication techniques because patterned nanostructures are essential for SER(R)S enhancement. Nanopatterned metallic structures or bulky prism-coupled thin metal films are commonly utilized for the excitation of plasmonic resonance[20], especially for surface plasmon polariton (SPP). Moreover, SPP is highly polarization and incident angle dependent. Scalable substrates are required to practically realize SER(R)S-based biosensors for real-time point-of-care (POC) applications. One solution is to use a Tamm plasmon polariton (TPP) cavity[21–23], where the thin metal film is coated with a distributed Bragg reflector (DBR). Here, the DBR provides the required optical phase matching to excite the plasmonic resonance (TPP) from the metal film. The TPP resonance provides high quality factor (narrow linewidth) mode, and it can be excited from normal incidence to a wide-angle range using both *p*-and *s*-polarizations of incident light[21]. More importantly, only thin film deposition technique is required to develop TPP cavity, thus wafer-scale fabrication at a low cost is possible. Due to these advantages, TPP cavities have been proposed for various applications such as perfect absorbers[24], hot electron photodetectors[25], thermal emitters[26], refractive index sensing[27,28], and lasers[29]. Most of the demonstrated TPP cavities are passive, which shows a single TPP resonance that is fixed for a particular wavelength. An attempt to realize tunable TPP resonance is demonstrated by integrating a passive TPP cavity with liquid crystals (LC)[30,31]. However, these systems provide very small resonance wavelength tunability and the cavities are thick due to the LC layer. Recently, a multi-resonant TPP cavity has been demonstrated in the mid-infrared (MIR) spectral band by combining an aperiodic DBR with a tunable cadmium oxide (CdO) layer for wavelength selective thermal emission[32]. This system also shows small TPP resonance tunability in the MIR frequencies due to the underlying mechanism of changing the carrier concentration of CdO. To realize large resonance tunability, it is necessary to tune the photonic bandgap (PBG) of DBR since the TPP resonance is excited within the PBG. It is also noted that TPP cavities are not yet proposed for SER(R)S applications.

Here, we demonstrate large TPP resonance tunability in the near-infrared (NIR) spectral band by tuning the PBG of DBR. To realize PBG tunability, we develop an ultralow-loss chalcogenide phase change material (PCM) based tunable DBR, where the PBG spectral band can be tuned by switching the structural phase of PCM from amorphous to crystalline. We show the excitation of narrow linewidth TPP modes by depositing a thin metal layer on top and bottom of the tunable DBR and demonstrate continuous tuning of TPP resonance by direct annealing and electrical heating. As one of the potential applications of the tunable TPP cavity, we demonstrate the SERRS with enhanced sensitivity by matching the second-order TPP resonance of the scalable Tamm cavity with the absorption peak of Raman reporter molecule at 785 nm excitation wavelength. We also performed a proof-of-concept biosensing application using the TPP cavity by detecting one of the cardiovascular disease (CVD) biomarkers such as cardiac Troponin I (cTnI) protein at biologically relevant concentrations.

## Results

### Development of tunable distributed Bragg reflector

To realize tunable DBR operating at optical frequencies, we choose an ultralow-loss PCM stibnite[33] ($Sb_2S_3$) as the constructing and active layer of the DBR. Chalcogenide-based PCMs are emerging as a potential platform for achieving diverse functionalities in active and reconfigurable nanophotonic devices[34]. PCMs offer stable and power-efficient means of tunability and reconfigurability for a wide spectral band from ultraviolet to terahertz[35–39]. Among PCMs, $Sb_2S_3$ shows lossless response in the wavelength >550 nm and >900 nm in amorphous and crystalline phase, respectively (see Supplementary Fig. 1a). Also note that the loss in $Sb_2S_3$ slightly increases with increase in annealing temperature from the amorphous phase (see Supplementary Fig. 1b). Here, we fabricated a tunable DBR by depositing an appropriate number of alternating layers of $Sb_2S_3$ and $SiO_2$ thin films on a quartz substrate (see Methods). Figure 1a shows the scanning electron microscope (SEM) image of fabricated $Sb_2S_3$-$SiO_2$ DBR, which consists of ten alternating layers of $Sb_2S_3$ and $SiO_2$ with thickness of $Sb_2S_3$ and $SiO_2$ layer is ~170 nm and ~100 nm, respectively.

By using transfer matrix method[40] (TMM), the calculated reflection spectrum of DBR at normal incidence is shown in Fig. 1b. As can be seen, when $Sb_2S_3$ is in an amorphous phase, DBR provides two PBGs, first order in the NIR wavelength (between 1100 nm and 1700 nm), and the second order in the visible wavelength (between 600 nm and 800 nm). The bandwidth of first-order PBG is almost double that of the second order. This is because the frequency bandwidth ($\Delta f_0$) of PBG is directly proportional to the central frequency ($f_0$) of PBG, $\Delta f_0 = (\frac{4f_0}{\pi})\sin^{-1}(\frac{n_2 - n_1}{n_2 + n_1})$, where $n_1$ and $n_2$ is the refractive index of dielectric layers. It is important to note that almost 100% reflection is obtained within the bandgaps by only using five bilayers of $Sb_2S_3$ and $SiO_2$. This is due to the high refractive index contrast ($\Delta n > 1.5$) between $Sb_2S_3$ and $SiO_2$ layers and the lossless feature of the amorphous $Sb_2S_3$ at >550 nm wavelength. Conventional passive DBR cavities of high (e.g., TiO2) and low (e.g., SiO2) refractive index dielectric materials require larger layers to achieve 100% reflection in the optical wavelength due to low refractive index contrast. The most important feature of the fabricated DBR is that large spectral bandgap tunability is possible by switching the structural phase of $Sb_2S_3$ layers in the DBR from amorphous to crystalline (see Methods). With the phase change of $Sb_2S_3$, the reflected intensity in the visible wavelength bandgap is drastically reduced due to the absorption of crystalline $Sb_2S_3$ layers, whereas the reflected intensity in the NIR wavelength bandgap is equal to that in amorphous state. In Fig. 1c, we show the measured reflection spectrum of DBR at normal incidence for both amorphous and crystalline phases of $Sb_2S_3$. The experimental results show a good agreement with the calculated results.

We further investigated the influence of incident angle and polarization of light on the PBG of tunable DBR. In Fig. 1d–g, we show the calculated and measured angular reflection spectra of DBR for both *p*-and *s*-polarizations when $Sb_2S_3$ is in an amorphous phase. For *p*-polarization, the photonic bandgap shrinks with increasing incident angle due to the Brewster effect at the interface between $SiO_2$ and $Sb_2S_3$ layers. However, the angle-independent response is observed for *s*-polarization. The measured incident angle and polarization response to the photonic bandgap of crystalline DBR are shown in Supplementary Fig. 2, which also follows the same characteristic. The measured and calculated transmission spectra also show the tunable bandgap feature of DBR (see Supplementary Fig. 3). In brief, the fabricated DBR provides tunable dual photonic bandgaps in optical wavelengths.

### Excitation and continuous tuning of TPP modes

Since TPP modes can be directly excited from free space, two scalable configurations can be used for this purpose, such as (i) a thin metal layer on top of the DBR (air-metal-DBR) and (ii) a thin metal layer

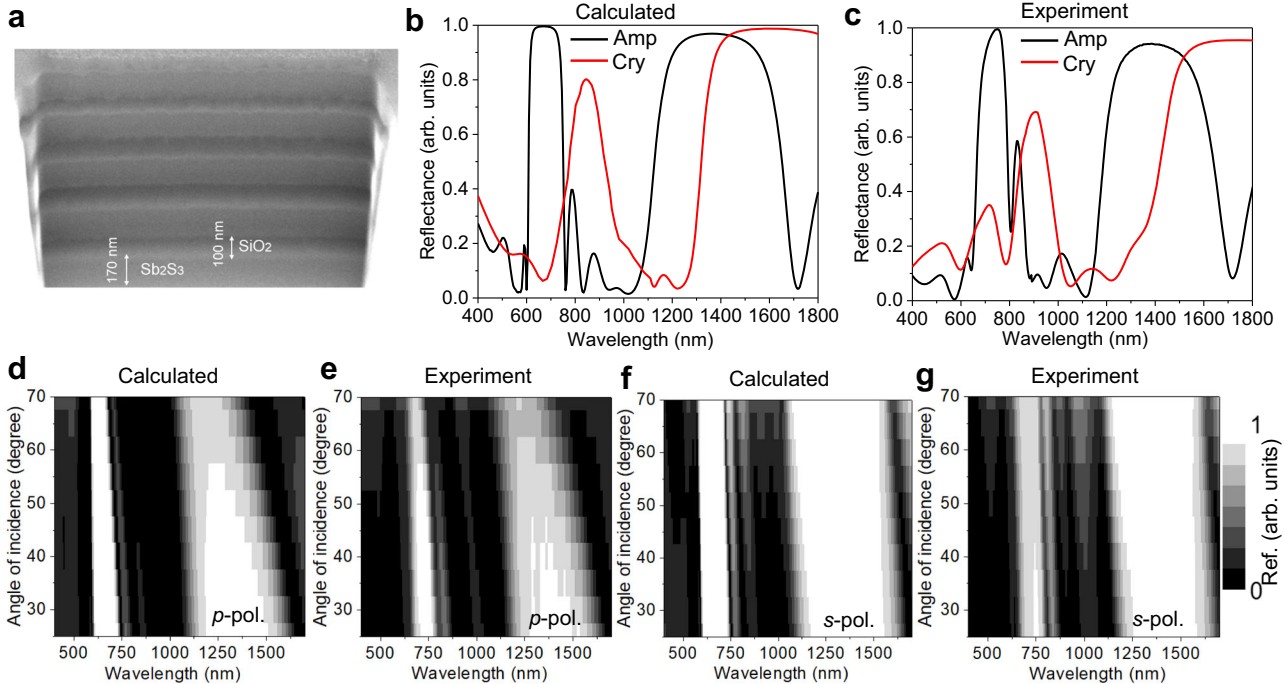

**Fig. 1 | Fabrication and characterization of tunable Sb₂S₃-SiO₂ DBR. a** SEM image of fabricated five bilayers of Sb₂S₃-SiO₂ DBR. The reflection spectrum of DBR at normal incidence for both amorphous (Amp) and crystalline (Cry) phases of Sb₂S₃ (**b**) Calculated and (**c**) Measured. Two-dimensional (2D) maps of angular reflection spectra of DBR are shown in (**d**) to (**g**). For *p*-polarizations (**d**) Calculated and (**e**) Measured. For *s*-polarization (**f**) Calculated and (**g**) Measured.

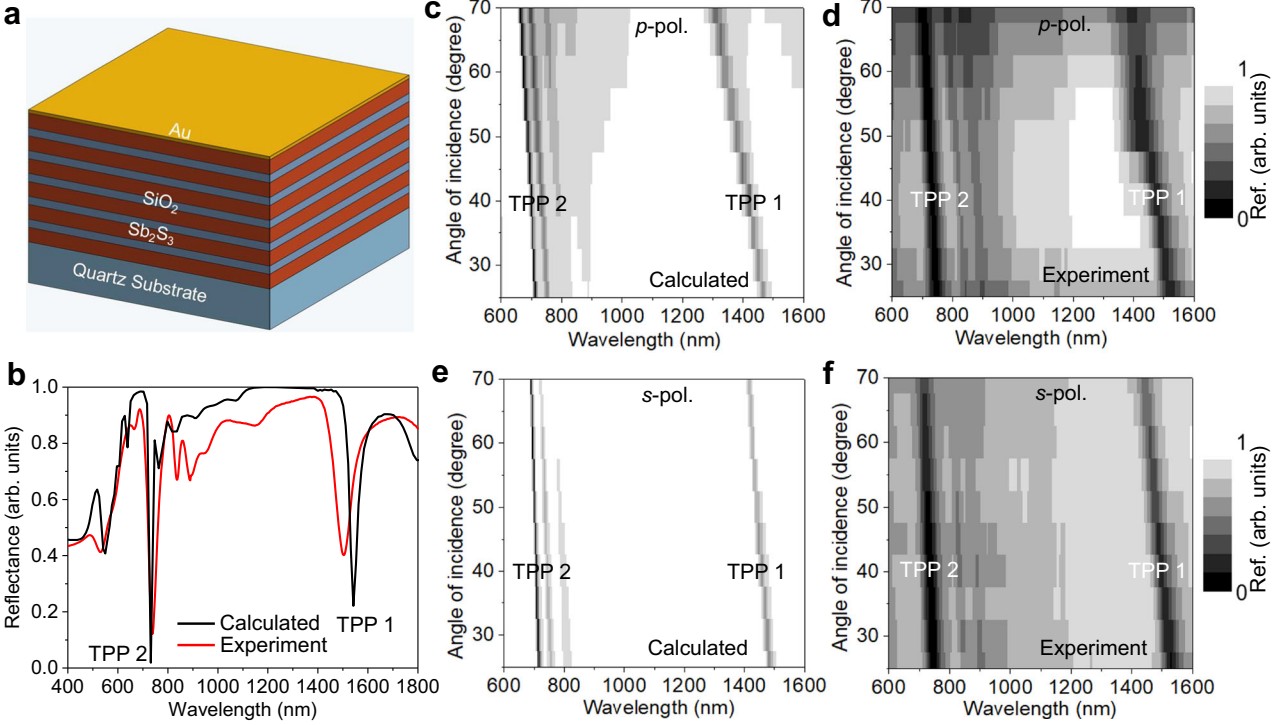

**Fig. 2 | Excitation of TPP modes of Au-coated DBR (TPP cavity). a** Schematic of fabricated TPP cavity. **b** Calculated and measured reflection spectra at normal incidence. 2D maps of angular reflection spectra of the TPP cavity are shown in (**c**) to (**f**). For *p*-polarization (**c**) calculated and (**d**) measured. For *s*-polarization (**e**) calculated and (**f**) measured.

between DBR and substrate (DBR-metal-substrate). For sensing applications, air-metal-DBR configuration is preferred because the analytes can directly experience the decaying near field on the metal layer. In Fig. 2a, we illustrate the schematic of the fabricated TPP cavity, where a

20-nm-thick gold layer was deposited on the Sb₂S₃ termination layer of DBR. The excitation of TPP mode is recognized by a narrow dip in the reflection spectrum of the TPP cavity and the dip lies within the bandgap of DBR. Figure 2b shows the measured and calculated

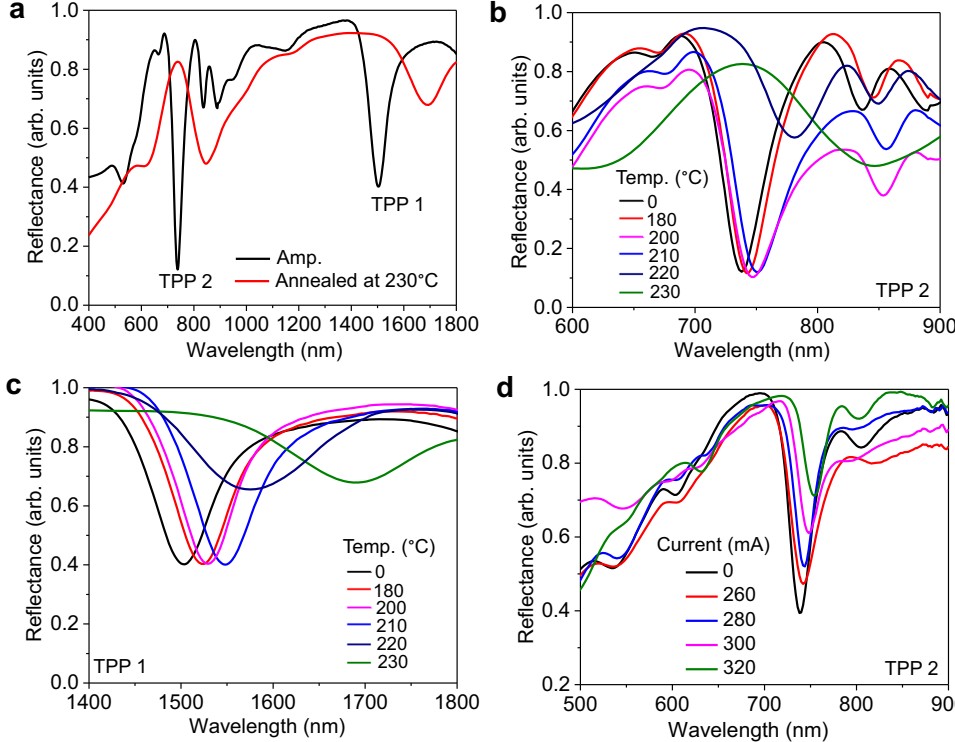

**Fig. 3 | Experimental demonstration of continuous tuning of TPP modes by direct annealing and electrically. a** Measured reflection spectra of TPP cavity when Sb$_2$S$_3$ is in Amp phase and annealed at 230 °C. Continuous tuning of TPP resonance wavelength with an increase in temperature for (**b**) second-order TPP and (**c**) first-order TPP. **d** Electrical continuous forward tuning of second order TPP mode using a microheater integrated TPP cavity.

reflection spectrum of the TPP cavity at normal incidence. As expected, two narrow linewidth TPP modes are excited at the resonance wavelength of 738 nm and 1504 nm, which lie within the visible (second order) and NIR (first order) PBGs of DBR, respectively. We further simulated the intensity field distribution of the TPP cavity at TPP 1 and TPP 2 resonance wavelengths and found that the field intensity is tightly confined at the DBR/Au interface and decays in both Au and DBR, which is a typical characteristic of a TPP mode (Supplementary Fig. 4).

The measured and calculated dispersion characteristics of the TPP cavity for both *p*-and *s*-polarizations are shown in Fig. 2c–f. For both polarizations, TPP modes spectrally overlap at normal incidence and spectrally separate with increasing angle of incidence. Both TPP modes blue shifted with the increase in incident angle for both polarizations. Also noted that the TPP mode for *p*-polarization blue shifted significantly than *s*-polarization at higher angles of incidence due to the bandgap shrinkage of DBR for *p*-polarization. We also fabricated a DBR-metal-substrate TPP cavity (see Supplementary Fig. 5a). The measured and calculated reflection spectra of this TPP cavity also show that TPP modes are excited within the bandgaps of DBR (Supplementary Fig. 5).

Since large spectral bandgap tunability is possible for the fabricated DBR, the TPP resonance wavelength can be continuously tuned in the forward direction by annealing the TPP cavity at different temperatures below the crystallization temperature of Sb$_2$S$_3$ (~280 °C). The calculated maximum resonance wavelength shifts of TPP 2 and TPP 1 by switching the phase of Sb$_2$S$_3$ from amorphous to crystalline is 153 nm and 295 nm, respectively (see Supplementary Fig. 6). Figure 3a shows the measured reflection spectrum of as-deposited (Amp) and annealed (at 230 °C) TPP cavity at normal incidence. The measured wavelength shift is 108 nm and 186 nm for TPP 2 and TPP 1 respectively. These values are lower than the calculated values because the optical constants of crystalline Sb$_2$S$_3$ were used in the calculation to obtain the

maximum wavelength shift. Note that TPP modes disappear when the sample is annealed above 230 °C due to the increased surface roughness of the film. The continuous resonance wavelength tuning of TPP 2 and TPP 1 mode with the increase in temperature (0 to 230 °C) is shown in Fig. 3b, c, respectively. A significant shift in resonance wavelength is obtained when the sample is annealed above 210 °C. This is because the refractive index of Sb$_2$S$_3$ layers drastically increases when the annealing temperature approaches the crystallization temperature of Sb$_2$S$_3$.

We demonstrate electrically continuous forward (amorphous to crystalline) tuning of TPP resonance using a microheater-integrated TPP cavity (see Supplementary Fig. 7a). Using the Joule heating mechanism, the whole sample was annealed by electric current-induced heating. The microheater was initially fabricated on a silicon substrate and followed by thin film deposition for the TPP cavity (see Methods). The temperature calibration of the microheater-integrated device was performed and found a linear variation of temperature (room temperature to 200 °C) with applied DC current (see Supplementary Fig. 7b). Figure 3d shows the electrical forward tuning of second order TPP mode with applied current. Since the linewidth of the microheater bar is 25 μm, a sample domain size of 25 μm × 25 μm was used for the reflectance measurement. At normal incidence, continuous tuning of TPP resonance towards longer wavelengths with increasing current (0 to 320 mA) is obtained. The obtained maximum wavelength shift is ~20 nm, which is the same as that obtained with direct annealing at 200 °C. We also confirmed that the temperature distribution is uniform over a sample area of 25 μm × 100 μm by monitoring the color change of DBR with applied current (Supplementary Note 1 and Supplementary Fig. 8). However, the development of an appropriate microheater[41] is required to achieve uniform higher temperature distribution on a scalable TPP cavity to realize large-area electrically tunable TPP resonances. In short, the resonance wavelength can be precisely tuned by properly selecting an external

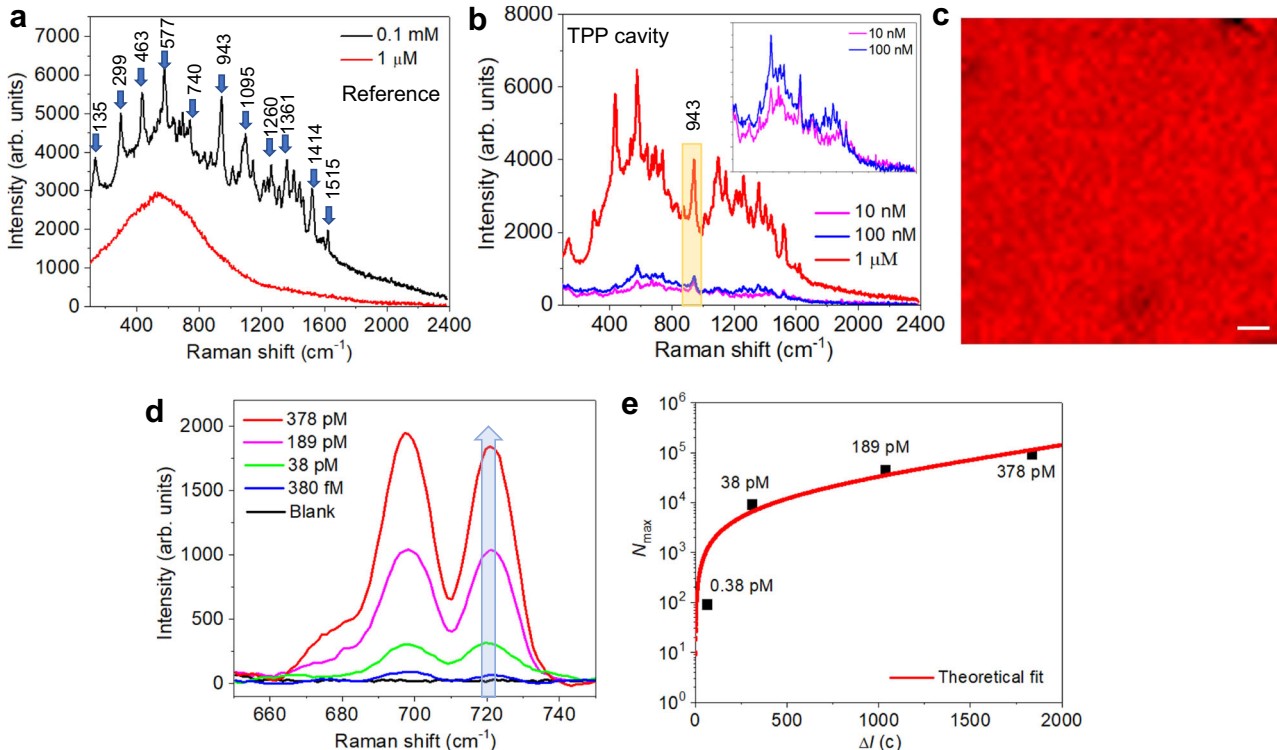

**Fig. 4 | Experimental demonstration of SERRS and protein sensing. a** Measured SERS spectrum of reference cavity using 0.1 mM and 1 μM cyanine dye concentrations. **b** Measured SERRS spectrum of TPP cavity using 1 μM, 100 nM, and 10 nM cyanine dye concentrations. Zoomed SERRS spectrum of 100 nM and 10 nM concentrations are shown in the inset of **b**. All SER(R)S spectra were subtracted from the background (dark) spectra. **c** Intensity mapping of 943 cm⁻¹ peak from cy7.5 dye, total mapping area is 120 μm × 120 μm. The scale bar is 12 μm. **d** SERRS spectra of the representative Raman band at 698 cm⁻¹ show an increase in SERRS intensity with an increase in the concentration of cTnI protein. **e** The maximum number of cTnI molecules ($N_{max}$) adsorbed in the illuminated sensor area versus the corresponding Raman intensity shift ($\Delta I$) with respect to a blank sample for different cTnI concentrations. The red curve represents the theoretical fit. Here, $\Delta I$ is calculated using 720 cm⁻¹ Raman band, as highlighted in **d**.

stimulus such as temperature or electric current, which is important for SERRS applications.

**Experimental demonstration of SERRS and protein detection**

Generally, nanopatterned metallic structures are required for SER(R)S to gain a large Raman scattering cross-section through electric field enhancement. Here, we reveal that the fabricated planar TPP cavity can be used as a potential scalable substrate for SERRS. We exploit the two features of the TPP cavity to achieve this goal: (i) since $Sb_2S_3$ is a soft material, the surface roughness of the Au thin film (~10 nm) deposited on $Sb_2S_3$ termination layer is high and it further increases with annealing and, (ii) at the TPP excitation wavelength, huge electromagnetic field confinement is achievable at the rough surface of $Sb_2S_3$/Au due to the high refractive index of $Sb_2S_3$. To avoid the direct laser exposure on soft $Sb_2S_3$ top layer, we first deposited a ~ 3 nm thick high refractive index oxide layer such as $BaTiO_3$ (BTO) on $Sb_2S_3$ termination layer of DBR and then deposited a thin Au layer of ~10 nm, followed by annealing the sample at 210 °C to obtain the first-order TPP resonance at 785 nm wavelength (see Supplementary Fig. 9). Using this approach, the surface roughness of the TPP cavity is greatly enhanced (see Supplementary Fig. 10). As a result, the minimum intensity at the TPP resonance wavelength (785 nm) is slightly increased. To compare the results, we fabricated a reference cavity consisting of quartz/$Sb_2S_3$ (~170 nm)/BTO (~3 nm)/Au (~10 nm) and annealed at 210 °C. Therefore, the surface roughness of TPP and reference cavity is almost the same ($R_a$ = 8.4 nm for ~10 nm coated Au film), however, the reference cavity cannot excite TPP mode (see Supplementary Fig. 11).

The SERRS experiments were performed using a NIR dye (cyanine 7.5 (cy7.5)), which has an absorption peak wavelength of around

785 nm (see Supplementary Fig. 12). Different concentrations of cy7.5 dye solutions were prepared and coated on TPP and reference cavities (see Methods). We used a micro-Raman system with a laser excitation wavelength of 785 nm (see Methods), which matches with TPP resonance wavelength and the absorption peak wavelength of cy7.5 solutions. Figure 4a shows the measured SERS spectrum of the reference cavity for two concentrations of cy7.5 dye solution. For a concentration of 0.1 mM, the prominent Raman signals of cy7.5 dye molecules are clearly visible due to the high surface roughness of the metallic film, however, the reference cavity is not sensitive enough to detect a lower concentration of 1 μM. In Fig. 4b, we show the SERRS spectrum of the TPP cavity using lower concentrations (1 μM, 100 nM, and 10 nM) of cy7.5 dye. Enhanced Raman signals of dye molecules are detectable even for a low concentration of 10 nM. This is due to the exact matching of dye absorption maximum with TPP resonance wavelength, and the huge field confinement attained on the rough surface of $Sb_2S_3$/Au at 785 nm wavelength (see Supplementary Fig. 13). It is worth mentioning that a similar kind of field enhancement mechanism was utilized for phase-sensitive refractive index sensing applications[42,43]. In contrast to the reference cavity, the Raman bands around 436 cm⁻¹ and 577 cm⁻¹ are selectively enhanced for the TPP cavity and no relative shift in the position of Raman bands is observed. It reveals that enhanced sensitivity obtained for the TPP cavity is due to the resonant Raman effect at 785 nm wavelength. The observed SERRS spectrum is reproducible for different samples (Supplementary Fig. 14). Figure 4c shows the intensity mapping of 943 cm⁻¹ peak from cy7.5 dye over an area of 120 μm × 120 μm, which shows uniform SERS enhancement. The high enhancement region is shown in bright red color and relatively lower enhancement regions are indicated as per

the color map. Overall, the substrates showed a high degree of reproducibility with a spatial variation in signal is less than 10%. SERS mapping demonstrated that the TPP substrate exhibits a robust signal enhancement due to the innovative tunable SERRS technology (for more details see Supplementary Note 2).

We also performed proof-of-concept biosensing by detecting cardiovascular disease (CVD) biomarkers. CVD is a major cause of mortality and morbidity worldwide. Accurate and timely diagnosis is crucial for effective management of CVD. Cardiac Troponin I (cTnI) is a protein found in cardiac muscle that has been extensively studied as a biomarker for CVD diagnosis due to its high specificity and sensitivity[44–46]. Hence cTnI is a valuable biomarker for the diagnosis and management of CVD. To evaluate the performance of the fabricated TPP cavity for protein sensing, cyanine 7 (cy7) was utilized as the Raman-active molecule. The absorption and SERS spectra of cy7 are shown in Supplementary Figs. 15 and 16, respectively. In Supplementary Fig. 17, we show the SERRS spectra from the cy7 corresponding to various biologically relevant cTnI cardiac biomarker protein concentrations ranging from 1.9 nM to 380 fM. We performed a proof-of-concept study in which we spiked thiolated cTnI proteins into a solution of 0.5 mg/mL BSA (Bovine Serum Albumin) (see Methods). We prepared these different spiked concentrations (1.9 nM to 380 fM) of cTnI into the sample vial containing BSA.

Figure 4d illustrates the spectra of different concentrations (378 pM to 380 fM) of cTnI protein and the representative Raman band at 698 cm$^{-1}$ increases in SERS signal intensity with an increase in cTnI protein concentration. We also performed SERS measurements of cTnI proteins alone by adsorbing on TPP substrate and could not observe any signal because the absorption of cTnI is negligible at 785 nm wavelength to yield SERRS enhancement (Supplementary Figs. 18 and 15). The sensitivity of the Raman intensity shift ($\Delta I(c)$) to the number of cTnI molecules ($N_{max}$) adsorbed on the illuminated sensor area is estimated using a previously proposed method[47,48] (for more details see Supplementary Note 3). In Fig. 4e, we plot $N_{max}$ versus $\Delta I(c)$ for the measured values of concentration from 380 fM to 378 pM, corresponding to $N_{max}$ ranging from 91 to 90720 molecules. The relationship between $N_{max}$ and $\Delta I(c)$ is nonlinear, which is consistent with a fitting function (red curve). The observed nonlinearity is due to the possibility of multiple adsorbed molecules on the sensor leading to interference effects (when concentration increases) and hence, with each additional molecule having a decreasing impact on the Raman intensity shift. This result shows that our SERRS-based biosensing platform is highly sensitive and accurate in measuring cTnI protein concentrations within the tested range of 378 pM to 380 fM from cTnI spiked plasma. The enhancement factor of TPP substrate is quantified using the well-known methodology[49,50] and the estimated enhancement factor is ~3.52 × 10$^7$ (see Supplementary Note 4). Based on this proof-of-concept study, the proposed scalable SERRS platform can detect cTnI protein concentration as low as 380 fM, which is in a few molecules (<100) detection regime. We detected spiked cTnI by ELISA (enzyme-linked immunosorbent assay) test and found that the sensitivity of the TPP-SERRS platform for the protein detection is comparable with the gold standard ELISA method (Supplementary Note 5, Supplementary Fig. 19, and Supplementary Table 1). We found that the sensitivity of the TPP cavity is higher compared to the SERS experiment performed using Au colloids (see Supplementary Fig. 20). Also note the performance of the TPP cavity is comparable with the existing metallic nanostructures-based SERRS demonstrations (Supplementary Table 2). The sensitivity of the TPP cavity can be further enhanced by nanopatterning the top Au layer, which can further improve the limit of detection (Supplementary Note 6 and Supplementary Figs. 21 and 22).

## Discussion

Different TPP cavities including tunable cavities have been reported for various applications. However, a small resonance wavelength tuning range was previously demonstrated by integrating DBR with liquid crystals and the CdO plasmonic layer. In this work, we realized a large TPP resonance tunability by tuning the photonic bandgaps of PCM-based DBR. By switching the structural phase of Sb$_2$S$_3$ layers in the DBR from amorphous to crystalline, a large resonance wavelength shift of 153 nm and 295 nm was obtained for the second and first-order TPP modes, respectively. It is also demonstrated that the TPP resonance can be continuously tuned in the forward direction by annealing the cavity at different temperatures below the crystallization temperature of Sb$_2$S$_3$. Since electrically tunable devices are more feasible for practical applications of tunable TPP cavities, we reported our initial results on electrically continuous forward tuning of TPP resonance over a uniform switching area of 25 μm × 100 μm using a microheater-integrated TPP cavity. More importantly, electrical reversible switching of Sb$_2$S$_3$-based cavities is necessary for reconfigurable photonic device applications, however, the efficient switching of large areas of thick (>1 μm) PCM-based devices is a challenging task[51–53].

It is well known that nanopatterned metallic structures are commonly used for SER(R)S applications. We demonstrated that the proposed tunable TPP cavity can be used as a scalable sensing substrate for SERRS. We realized SERRS with enhanced sensitivity by matching the TPP resonance wavelength with the absorption peak wavelength of cyanine dye at 785 nm excitation wavelength. Using our approach, it is possible to clearly see all the Raman signals of cyanine dye molecules compared to the previously reported resonant Raman effect, where only the Raman excitation wavelength was matched with the absorption peak wavelength of cyanine dye[15]. We have demonstrated the biosensing application of the resultant substrate with the detection of clinically relevant concentrations of CVD biomarker cTnI. The development of highly sensitive detection methods has allowed for the detection of 380 fM to 378 pM concentrations of cTnI protein in spiked plasma samples with few molecules detection limit, which has important clinical implications. By employing the tunability of the substrate and the SERRS approach, it could be practically possible to detect a wide range of biomolecules in a label-free manner, particularly relevant for molecules that possess moderate innate Raman cross-section. In addition, the excitation of multiple TPP modes at different excitation wavelengths can be utilized to realize multiplexed detection (Supplementary Fig. 23), as such assay is essential in sensing applications because it can provide higher dimensional biochemical analysis of complex samples[54,55].

The sensitivity of the sensor can be further improved by engineering the morphology of the sensing surface. The spectral tunability of TPP mode can also be used to mitigate the fabrication error. The ease of fabrication using thin film deposition techniques, the optical excitation without using any coupling techniques, polarization and incident angle independent response, and high resonance quality factor modes with tunable features make the TPP cavity a potential scalable sensing platform for practical SERS applications. The proposed tunable TPP cavity can be used for other potential applications such as refractive index sensors, tunable narrowband absorbers and filters, and tunable lasers.

## Methods
### Sample fabrication
The DBR samples were fabricated by the sequential deposition of Sb$_2$S$_3$ and SiO$_2$ thin films on quartz and silicon substrates. Prior to deposition, the substrates were pre-cleaned using acetone, ethanol, and deionized water. Thin films of Sb$_2$S$_3$, BTO, and SiO$_2$ were deposited using RF magnetron sputtering. To achieve uniform deposition of thin films, the distance between the substrate and target was kept at 15 cm. Room temperature deposition under a high-purity argon (99.999%) atmosphere at a deposition pressure of 10 mTorr was carried out. To switch the structural phase of Sb$_2$S$_3$ layers in the DBR from amorphous

to crystalline, the samples were annealed at 250 °C on a hot plate for 15 min. For TPP cavities, the thin layer of Au was deposited using the thermal evaporation technique at a deposition rate of 0.1 A/s. In this case, samples were annealed at 230 °C on a hot plate for 5 min to avoid the increased surface roughness of the film.

Microheater fabrication and Temperature calibration: Using UV illumination-based photolithography, the metallic microheater was fabricated on a Si substrate. Tungsten (W) was used as the heater metal and 200 nm thick W film was deposited over the entire area using DC magnetron sputtering and followed by a lift-off process to leave behind the W only in the patterned area. The microheater was then subjected to temperature calibration using a source meter (Keithley 2450). Two probes were used to apply the DC current to the heating element and a thermocouple was mounted at the center of the microheater to monitor the temperature change with applied current. To fabricate microheater-integrated TPP cavities, the thin films of $Sb_2S_3$, $SiO_2$, and Au were directly deposited in the whole area of the sample.

2D nanohole grating fabrication: A focused (gallium)-ion beam (FEI Helios NanoLab 600) was used to fabricate 2D nanohole gratings. By using FIB with a beam current ≤18 pA and dosage of 10 mC/cm², we milled on 30 nm thick Au thin film deposited on the $Sb_2S_3$ termination layer of DBR. The patterned grating area was 30 μm x 30 μm. A scanning electron microscope (FEI Helios NanoLab 600) was used for imaging the cross-section and surface morphology of the samples. The surface roughness of the samples was determined using an atomic force microscope (Burker).

### Ellipsometry characterizations
Variable-angle high-resolution spectroscopic ellipsometry (J.A. Woollam Co., Inc., V-VASE) was used to determine the thickness and the optical constants of $Sb_2S_3$, Au, and $SiO_2$ thin films.

### Spectroscopy measurements
The normal incidence reflectance measurements were performed using a microspectrophotometer (Jasco, MSV-5200) with a sampling domain size of 100 μm × 100 μm and 25 μm × 25 μm for scalable and nanopatterned TPP cavities, respectively. The reflection measurement of microheater-integrated samples was performed using a sampling domain size of 25 μm × 25 μm. Angular reflection measurements of scalable TPP samples were performed using a variable angle spectroscopic ellipsometry with a beam spot size of 2 mm × 2 mm (J.A. Woollam Co., Inc., V-VASE). The normal incidence transmission measurements of scalable TPP samples were performed using the same ellipsometer.

### Raman measurements
SER(R)S measurements of cy7.5 dye were performed using a uRaman 785 system (EiNST), where the excitation source wavelength was 785 nm. A 20× Plan APO objective lens with $NA = 0.75$ (Nikon) was used to focus the laser beam. Raman signals were collected under an excitation power = 0.5 mW, acquisition time = 5 seconds, and averaging = 1. All the SER(R)S spectra were subtracted from the dark spectra.

For Raman mapping of the samples and protein sensing, a Renishaw InVia Raman upright microscope equipped with a 785 nm laser was employed. This instrument was coupled with a Leica microscope, and the laser light was focused onto the sample using an objective lens with a ×50 magnification and $NA = 0.5$. The scattered Raman signal was collected through the same lens, and prominent Rayleigh scattering was blocked using a notch filter. The beam spot size on the sample was carefully controlled to be approximately 2 μm to ensure high spatial resolution. To ensure statistical significance, over 10 different areas were analyzed on each sample, and 10 spectra were acquired from each area. To acquire each spectrum, the laser was focused on the sample for 10 seconds, and the Raman signal was

integrated over the range of 600–1800 cm⁻¹. To analyze the acquired spectra, WiRE™ v3.4 software was employed. The fluorescence background was subtracted using cubic spline interpolation, and the instrument was calibrated using a standard silicon with its 520 cm⁻¹ Raman peak.

### Cyanine dye solution preparation
A cyanine family dye, Cy7.5 NHS ester (non-sulfonated) with an absorption maximum wavelength of 788 nm and emission peak wavelength of 808 nm was purchased from Lumiprobe. Cyanine dye solutions with varying concentrations from 0.1 mM to 10 nM were prepared by serial dilution in DMSO solvent. Samples were dipped in dye solutions for a short time and then performed the Raman measurements. The absorption spectrum of the dye solution was determined using a UV-Vis Spectrophotometer.

### Protein sensing
2-Iminothiolane hydrochloride (Traut's reagent) (Sigma Aldrich), recombinant human cardiac Troponin I protein (cTnI) (Abcam), Anti-cardiac Troponin I antibody (Anti- cTnI) (Abcam), PE/Cy7® Conjugation Kit - Lightning-Link® (ab102903) (Abcam), bovine serum albumin (BSA) (Sigma Aldrich), ethanol (Merck) and phosphate-buffered saline (PBS) (Lonza) were purchased as indicated.

Commercial preparations of cTnI proteins and antibodies were used without modification. To evaluate the performance of the fabricated substrate, cyanine 7 (cy7) was utilized as the Raman-active molecule. cTnI protein (Abcam) was treated with 14 mM Traut's reagent, and the resultant mixture was filtered using an Ultra-0.5 Centrifugal Filter Unit (Amicon) to eliminate excess reagent. Lightning-link conjugation kit (Abcam) was used to conjugate the cy7 fluorescent probe onto the cTnI antibody and the mixture was incubated for a short time. The resulting solution was filtered to remove excess unreacted reagents, and the cy7-tagged antibody filtrate was reconstituted back to 0.1 mg/mL. To confirm the specificity of our biosensing methodology, we have performed a negative control in which we used another protein (matrix metalloprotein) instead of cTnI. Based on the results we found that when non-cTnI protein is used the cy7-tagged anti-cTnI did not bind to the substrate and expressed a weak background signal which is lower than the lowest concentration of Troponin I tested.

### Numerical simulations
All the reflection and transmission spectra were simulated using a transfer matrix-based simulation model written in MATLAB. The finite difference time domain (FDTD) was used to simulate the electric field intensity distribution. The commercially available Ansys Lumerical FDTD software was used for this purpose. In the 3D simulation, the periodic boundary condition was used along the $x$ and $y$ directions, and a perfectly matched layer (PML) boundary condition was used along the $z$ direction. Experimentally determined spectral optical constants of $Sb_2S_3$, BTO, Au, and $SiO_2$ were used in the simulations.

### Reporting summary
Further information on research design is available in the Nature Portfolio Reporting Summary linked to this article.

## Data availability
Data used in this study are available from the corresponding authors upon request. The experimental and simulation data are provided in the Source Data file. Source data are provided with this paper.

## Code availability
The codes used in this paper are available from the corresponding authors upon request.

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

## Acknowledgements
J.T. and K.V.S. acknowledge funding support from the National Research Foundation Singapore under the CRP program (Grant No. NRF-CRP26-2021-0004), A*STAR (Agency for Science, Technology and Research) under AME IRG Program (Grant No. A2083c0058 and A20E5c0084) and HBMS IAF-PP (Grant No. H19H6a0025). U.S.D., J.P., and M.O. acknowledge the following A*STAR funding support: IAF-PP Grant H19H6a0025 and BMRC UIBR Grant. R.S. acknowledges the funding support from the National Research Foundation Singapore (Award No.: NRF-CRP23-2019-0005). The authors also would like to thank Jodie, Foo Chuan Yue, and Ran, Chua Zhi Tong for their help in SERS measurements.

## Author contributions
J.T. initiated the tunable surface-enhanced Raman spectroscopy work. K.V.S. developed the approach, designed the research, fabricated, and characterized the samples, performed spectroscopic and Raman measurements, carried out calculations and numerical simulations, and wrote the manuscript. J.P. and U.S.D. designed the biosensing experiments, performed Raman measurements, conducted ELISA tests, carried out calculations, and wrote a part of the manuscript. P.P. fabricated and characterized samples. Y.L. performed AFM characterization. J.T., M.O., and R.S. supervised the project. All authors analyzed the data and discussed the results.

## Competing interests
The authors declare no competing interests.
