## [Peer Review File · Nature Communications]

REVIEWER COMMENTS

Reviewer #1 (Remarks to the Author):

The authors demonstrate a tunable Tamm plasmon polariton (TPP) device by using a phase change material (PCM, Sb₂S₃). And, they tune the TPP resonance peak to the specific wavelength, ~ 785 nm, for carrying out the Surface enhanced Resonance Raman spectroscopy (SERRS) and applying SERRS for sensing cardiac Troponin I protein (cTnI). The results are impressive. The sensitivity of the proposed TPP device can detect cTnI protein concentrations to ~380 fM. Here, I have some questions and comments may help the authors to improve the manuscript.

1. They claim the applied Sb₂S₃ is a low loss material. However, the Sb₂S₃ becomes a loss material when they try to change the crystallinity of Sb₂S₃ for tuning the TPP resonance wavelength to ~ 785 nm (Fig. S1). It turns out that the TPP at ~785 nm has a low resonance quality factor (Fig. 3b). Please estimate the enhancement of electric field at 785 nm. Then the readers can know the enhanced sensitivity for detection is reasonable or not.
2. The electric field is mainly confined between the metallic layer and the DBR for a TPP device. The detected materials, cy7.5 dye, cannot access the confinement region directly. Why do they observe a big difference of SERRS between the reference and the proposed TPP cavities in Fig. 4a and Fig. 4b? Nanopatterned TPP cavity in Fig. S15 can solve the abovementioned problems. However, the nanopatterned TPP cavity cannot be scaled up. Therefore, it is out of the scope in this work.
3. What happened if the proposed TPP substrate is applied to detect cTnI directly rather than cy7 tagged-cTnI in Fig. 4d? What are the absorption spectra of cy7 tagged-cTnI and cTnI?
4. The proposed structure shows two resonance Tamm plasmon polariton (TPP) peaks from two photonic band gaps (PBG). The authors name that the first peak in the visible range is the "first order". Please check it according to the conventional notation, e.g. Bragg's law. The authors may provide a simple description for the different bandwidths between the visible and near infrared ranges.
5. The authors have studied the influence of incident angle and polarization of light (Fig. 1 and Fig. S2). They might discuss why they apply the normal incident rather than other angles in their sensing applications. The color in Fig. S2(a) seems not right.
6. In line 177, they claim that Fig. S4 shows the measured and calculated TPP modes. However, Fig. S4 only shows the calculated results.
7. In line 190, Fig. S5 should be replaced by Fig. S4.
8. In line 234, the sample is annealed at 210°C to obtain the first TPP resonance at 785 nm wavelength. It is not consistent with Fig 3(b). In Fig. 3(b), the annealing temperature for 785 nm TPP is ~220°C.
9. What's the ratio of confined electric field between the TPP and reference cavities (Fig. S8 and Fig. S9)?

Reviewer #2 (Remarks to the Author):

I cannot recommend the publication of this manuscript in Nat. Comm. because the originality and novelty of the study reported are insufficient to justify the publication in it. This study is rather technical report not so much scientific. The idea of a tunable Tamm plasmon polariton (TPP) cavity

based on a distributed Bragg reflector (DBR) on metal as a scalable sensing platform is interesting, but I do not think this technique will be used frequently in SERRS community. The authors reported "The development of a dynamically tunable plasmonic system would enable the precise matching of achieving SERRS by exactly matching the resonance wavelength with the absorption peak wavelength of biomolecules, thereby improving the sensitivity and the selectivity of SERRS." However, usually in the case of SERRS the exact matching of the resonance wavelength with the absorption peak wavelength of a molecular is not always necessary. Moreover, the authors showed only one example of SERS (Figure 4) in the text, and they did not give enough interpretation. Therefore, I do not think they succeeded in demonstrate their method in measuring highly sensitive SERRS.

I would suggest the authors to submit their results to a more specialised journal.

Reviewer #3 (Remarks to the Author):

Major revision

The authors proposed a tunable Tamm plasmon polariton (TPP) cavity based on a distributed Bragg reflector (DBR) on metal as a scalable sensing platform for SERRS. The phase change materials used in the DBR allows for tuning of the operating wavelength. Furthermore, the authors demonstrated improved SERRS compared to reference measurements. Although the concept is novel and the demonstrations are solid, some major comments need to be addressed before further consideration. Major comments:

1. Using the TPP mode to enhance the SERRS is novel. However, the performance of the TPP devices in terms of SERRS is not quantified and benchmarked properly. Unless the performance of TPP device is better or comparable to other strategies (demonstrations), I do not think the work meets the bar of nature comm.
2. Based on my understanding of Phase change materials (PCMs), it is extremely challenging (if not impossible) to amorphized large PCM areas (~ 30 um square is the largest size I'm aware of, Abdollahramezani, Sajjad, et al. "Electrically driven reprogrammable phase-change metasurface reaching 80% efficiency." Nature communications 13.1 (2022): 1696.). Thus, the authors need to be careful about the wording in the discussion sections regarding the amorphization process.
3. Following the previous comment, in my opinion, adding the PCM is a great idea, allowing the device to be tuned or adjusted towards longer wavelength, so that fabrication error can be mitigated. The authors mentioned that the device used for SERRS was annealed at 210 C, is it to shift the resonance to compensate for fabrication error? I recommend the authors highlight this aspect.
4. As the authors suggested, the electric heating is not uniform. Please include thermal images or simulations to more quantitatively discuss: (1) how non-uniform the temperature is; (2) how does the non-uniform temperature affect the measurement (the sampling domain is only 100 by 100 um, does it only affect the reproductivity? Or it also makes the collected region non-uniform?)
5. The loss of Sb₂S₃ is quite high at ~ 785 nm. In Fig. 1 b and c, the first DBR reflection band is almost smashed after switching to crystalline state. Please comment on the loss from Sb₂S₃, how it changes the SERRS performance.

Minor comments:

1. Line 61, the authors stated that 10^6 enhancement by SERRS has been demonstrated, but the authors did not quantify the enhancement by the TPP structure.
2. Line 163, it is not immediately clear to readers why the sensing application requires the metal side contacted with the external environment. I recommend mentioning the field-profiles here.

We first provide a summary of revisions followed by detailed point-by-point summary of the reviewer's comments.

SUMMARY OF MAJOR REVISIONS

- 9 figures with additional clarification, theoretical calculations, simulations, and experimental results.
- 8 additional references in the main text.

In response to Reviewer #1, the key revisions include:

- Figures 1b, 9, 13, 15, and 18 in the supplementary information.
- Clarification and discussion in the main text regarding the loss of Sb_2S_3 and field distribution on the sensing surface.

In response to Reviewer #2, the key revisions include:

- Clarification and discussion in the main text regarding the novelty and potential applications of the proposed TPP cavity.
- Clarification and discussion on the importance of exactly matching the resonance wavelength with the absorption peak wavelength of biomolecules and its potential applications

In response to Reviewer #3, the key revisions include:

- Figures 1b, 8, 9, 13, 19, 20 and 23 in the supplementary information.
- Figure 4e in the revised text.
- Discussion on loss in Sb_2S_3 film, uniform temperature distribution on the switching area, quantification, benchmarking, and comparison with other strategies.

POINT-BY-POINT RESPONSE

Reviewer #1

The authors demonstrate a tunable Tamm plasmon polariton (TPP) device by using a phase change material (PCM, Sb₂S₃). And, they tune the TPP resonance peak to the specific wavelength, ~ 785 nm, for carrying out the Surface enhanced Resonance Raman spectroscopy (SERRS) and applying SERRS for sensing cardiac Troponin I protein (cTnI). The results are impressive. The sensitivity of the proposed TPP device can detect cTnI protein concentrations to ~380 fM. Here, I have some questions and comments that may help the authors to improve the manuscript.

Reply: We would like to thank the reviewer for the time and efforts dedicated in reviewing the manuscript and providing valuable feedback and comments. We have clarified and addressed all the points raised by the reviewer. Here, we have provided detailed responses to the comments.

Comment 1: *They claim the applied Sb₂S₃ is a low loss material. However, the Sb₂S₃ becomes a loss material when they try to change the crystallinity of Sb₂S₃ for tuning the TPP resonance wavelength to ~ 785 nm (Fig. S1). It turns out that the TPP at ~785 nm has a low resonance quality factor (Fig. 3b). Please estimate the enhancement of the electric field at 785 nm. Then the readers can know the enhanced sensitivity for detection is reasonable or not*

Reply: Thank you for this important comment. In supplementary Fig. 1, we presented the optical constants of amorphous ('as deposited') and crystalline (annealed) Sb₂S₃ thin films. Since the crystallization temperature of Sb₂S₃ is around 280°C (*Adv. Funt. Mater.* **29**, 1806181 (2019)), the sample was annealed at 290° C for crystallization. As pointed out by the reviewer, the crystalline Sb₂S₃ film is lossy at 785 nm ($k=0.105$). To investigate the loss of Sb₂S₃ at different annealing temperatures, we measured the extinction coefficient (k) of 170 nm thick Sb₂S₃ film annealed at temperature of 200°C, 210°C and 220°C and plotted along with k values of amorphous and crystalline films, as shown in Fig. R1. It is evident that the obtained k value at 785 nm wavelength

is negligible for the samples annealed at $\leq 220^\circ\text{C}$. The measured k value is 0.006 at 785 nm for the sample annealed at 220°C . Therefore, the observed low resonance quality factor of the mode shown in Fig. 3b is due to the surface roughness of the Au film coated TPP cavity.

Fig. R1 Measured extinction coefficient of Sb_2S_3 thin film at different temperatures

It is important to note that the TPP resonance wavelength blue shifts with increase in Au film thickness (Fig. R2a). Also, the intensity at the resonance wavelength and the quality factor of the mode increase with increasing Au film thickness. In Fig. 2 and 3 (manuscript text), we showed the results obtained using a 20 nm Au coated TPP cavity, which exhibited TPP resonance at 745 nm for ‘as deposited’ sample and at 785 nm after annealing at 220°C . **As stated in the manuscript, we used 10 nm Au film for SERRS studies.** The initial TPP resonance was at 760 nm for ‘as deposited’ sample and then shifted to 785 nm after annealing at 210°C (Fig. R2b). It is evident that the experimentally obtained resonance quality factor of the 10 nm Au coated TPP cavity annealed at 210°C is significant compared to the 20 nm Au coated TPP cavity annealed at 220°C . The reduced resonance quality factor of sample annealed at 210°C compared to ‘as deposited’ sample is due to the increased surface roughness of the TPP cavity even though the loss

of Sb_2S_3 is zero at 785 nm (Fig. R2b). Also note that the resonance quality factor of the 20 nm Au coated TPP cavity annealed at 210°C shows comparable to that of an amorphous sample (Fig.3b in the manuscript). It shows that the surface roughness increases with decreasing the Au thickness. This is the reason why we used an optimized thickness of 10 nm Au layer to obtain high surface roughness as well as a considerable resonance quality factor at 785 nm. Also note that this slightly decreased resonance quality factor of TPP resonance obtained in the far-field reflection measurement may not affect SERS because SERS is a near-field effect. As suggested by the reviewer, we have performed FDTD simulations to estimate the enhancement of electric field at 785 nm by using 10 nm Au film coated TPP cavity and the optical constants of Sb_2S_3 film that annealed at 210°C. The results are presented in Reply to Comment 2.

Fig. R2 (a) Calculated reflection spectrum of TPP cavity for different thicknesses of Au layer. (b) Measured reflection spectrum of 10 nm Au coated TPP cavity (as deposited and annealed at 210°C).

Figure R1 and Fig. R2b has been included in the supplementary information as Fig. 1b and Fig.9, respectively.

The following text has been included in the manuscript to address this point:

“Also note that the loss in Sb_2S_3 slightly increases with increase in annealing temperature from the amorphous phase (see Supplementary Fig.1b)” (Page 6)

“Using this approach, the surface roughness of the TPP cavity is greatly enhanced (see Supplementary Fig. 10). As a result, the minimum intensity at the TPP resonance wavelength (785 nm) is slightly increased” (Page 13)

Comment 2: *The electric field is mainly confined between the metallic layer and the DBR for a TPP device. The detected materials, cy7.5 dye, cannot access the confinement region directly. Why do they observe a big difference of SERRS between the reference and the proposed TPP cavities in Fig. 4a and Fig. 4b? Nanopatterned TPP cavity in Fig. S15 can solve the above-mentioned problems. However, the nanopatterned TPP cavity cannot be scaled up. Therefore, it is out of the scope in this work*

Reply: In TPP cavities, field intensity tightly confines at DBR/metal interface and decays exponentially to metal as well as to the DBR [*Phys. Rev. B* **76**, 165415 (2007)]. By utilizing the exponentially decaying field in the metal layer, TPP cavities have been widely employed for refractive index sensing applications [*Optics Express* **22**, 14524-14529 (2014)]. Similarly, as shown in supplementary Fig. 4, the field intensity is tightly confined at $\text{Sb}_2\text{S}_3/\text{Au}$ interface and decays in both Au and Sb_2S_3 layers, where the Au thin film thickness is 20 nm. Note that we used 10 nm Au film coated TPP cavities for SERS experiments. The topology of this TPP cavity looks like random nanostructures due to high surface roughness. Since biosensing is a surface sensing mechanism, the biomolecules attached to the $\text{Sb}_2\text{S}_3/\text{Au}$ random nanostructure surface directly experience this near field. However, it is not straightforward to simulate the electric field distribution of TPP cavity with 10 nm Au film due to random nanostructure surface. Since the topology of the reference sample is also the same and it is to confirm that the enhanced SERS

signal obtained for TPP cavity is due to the excitation of Tamm mode at 785 nm, we simulated the intensity field distribution by considering the top layer as 10 nm Au film.

As pointed out by the reviewer, we observed a large SERS enhancement difference between reference and TPP cavities. The observed SERS enhancement of the reference cavity is only due to the random nanostructure surface of the cavity. However, the large SERS enhancement of TPP cavity is due to the huge field confinement on the $\text{Sb}_2\text{S}_3/\text{Au}$ random nanostructure surface by TPP mode excitation at 785 nm. To emphasize it, here we compare the field distribution of TPP and reference cavities at 785 nm. Figure R3 illustrates the simulated intensity distributions of TPP, and reference cavities zoomed at $\text{Sb}_2\text{S}_3/\text{Au}$ interface. As shown in Fig. R3a, the field is tightly confined at $\text{Sb}_2\text{S}_3/\text{Au}$ interface and its intensity decays to the Au layer for TPP cavity. However, there is no considerable intensity confinement at $\text{Sb}_2\text{S}_3/\text{Au}$ interface for reference cavity (Fig. R3b). It is emphasized that the confined field distribution obtained for TPP cavity is due to the excitation of TPP at 785 nm. Figure R3c shows the ratio of the intensity distribution between TPP and reference cavities. It is evident that maximum intensity is confined within the Au layer. This is the reason why TPP cavity provides enhanced SERRS sensitivity compared to reference cavity.

It has been reported that improved limit of detection with single molecule sensing is possible by exactly matching the resonance wavelength with the absorption peak wavelength of molecules [*J. Am. Chem. Soc.* **134**, 1966–1969 (2012)] (also see Table 1 in the Supplementary Information). In all the existing works, ultrahigh sensitivity was achieved by combining SERRS with shorter wavelength laser excitation because the enhancement factor is directly proportional to the fourth power of the excitation frequency. In addition, nanopatterned and self-assembled nanostructures were used in all the reported SERRS works. It is important to note that our proposed TPP cavity

achieved femtomolar detection level using a near IR laser (785 nm) and scalable platform. As shown in Fig. 4a (manuscript), reference sample is unable to detect 1 μ M concentration of cy7.5 solution, where the laser excitation wavelength is only matched with the cy7.5 absorption peak wavelength. However, TPP cavity is capable of detecting 10 nM concentration of cy7.5. It shows that the limit of detection increases by exactly matching the resonance wavelength with the absorption peak wavelength of the molecule. Note that the enhanced sensitivity obtained for protein detection (compared to cy7.5 alone) is due to the specific binding of proteins on the sensor surface via surface functionalization. We have also estimated adsorbed cTnI molecules on the illuminated sensor area for different concentrations of cTnI (see Supplementary Note 2), which shows that TPP cavity operates in the few molecules detection regime.

Fig. R3 2D cross-sectional map of intensity field distribution at 785 nm, zoomed at Sb₂S₃/Au interface for (a) TPP cavity and (b) reference cavity. (c) Ratio of the intensity field distribution as a function of z-direction.

The following text has been included in the manuscript to address this point:

“We further simulated the intensity field distribution of TPP cavity at TPP 1 and TPP 2 resonance wavelengths and found that the field intensity is tightly confined at DBR/Au interface and decays

in both Au and DBR, which is a typical characteristic of TPP modes (see Supplementary Fig. 4)” (Page 9)

“This is due to the exact matching of dye absorption maximum with TPP resonance wavelength, and the huge field confinement attained on the rough surface of Sb₂S₃/Au at 785 nm wavelength (see Supplementary Fig. 13)” (Page 13)

Figure R3 has been included in the supplementary information as Fig. 13.

We agree with the reviewer that the nanopatterned TPP cavity is out of the scope in this work. The purpose of including nanopatterned TPP cavity in this work is to show that the TPP resonance can also be tuned by changing structural parameters of the nanopattern as well as by annealing the sample. Also, note that the scalable TPP cavity only shows enhanced performance when the thickness of Au film is ≤ 10 nm. As stated above, the proposed scalable sensing platform can detect few biomolecules. To fabricate high aspect ratio and uniform nanopatterns on the TPP cavity, an Au film with thickness ≥ 30 nm is required. An extra advantage of nanopatterned TPP cavity is that SERS signal can be further enhanced by properly designing the shape and size of the nanopatterns, which could further improve the limit of detection.

The respective text in the manuscript has been amended as,

“The sensitivity of TPP cavity can be further enhanced by nanopatterning the top Au layer, which can further improve the limit of detection (see Supplementary Note 5)” (Page 16)

Comment 3: *What happens if the proposed TPP substrate is applied to detect cTnI directly rather than cy7 tagged-cTnI in Fig. 4d? What are the absorption spectra of cy7 tagged-cTnI and cTnI?*

Reply: As suggested by reviewer, we have performed SERS measurements using cTnI alone and plotted along with cy7 tagged cTnI in Fig. R4. Figure R5 shows the measured absorption spectra

of cy7 tagged-cTnI and cTnI alone. As shown in Fig. R4, SERS intensity of cTnI alone is negligible compared to cy7 tagged-cTnI because cTnI protein did not show any absorption band in the visible wavelength (Fig. R5a). At the same time, cy7 tagged-cTnI shows a significant absorption around the excitation wavelength (785 nm). Additionally, we confirmed that the absorption peak of the cy7 is not altered significantly when it is attached to cTnI proteins.

Fig. R4 SERS spectrum of cTnI alone and cy7 tagged cTnI

Fig. R5 Absorption spectrum of (a) cTnI and (b) cy7 tagged cTnI

Figure R4 and Fig. R5 has been included in the supplementary information as Fig. 18 and Fig. 15, respectively.

The following text has been included in the manuscript to address this point,

“We also performed SERS measurements of cTnI proteins alone by adsorbing on TPP substrate and could not observe any signal because the absorption of cTnI is negligible at 785 nm wavelength to yield SERRS enhancement (see Supplementary Fig. 18 and Fig. 15)” (Page 15)

Comment 4: *The proposed structure shows two resonance Tamm plasmon polariton (TPP) peaks from two photonic band gaps (PBG). The authors name that the first peak in the visible range is the “first order”. Please check it according to the conventional notation, e.g. Bragg’s law. The authors may provide a simple description for the different bandwidths between the visible and near infrared ranges*

Reply: Thank you for pointing out the mistake in the nomenclature of the bandgaps. The obtained two bandgaps of Sb₂S₃-SiO₂ DBR can be explained based on Bragg's law, $n\lambda=2d\sin\theta$, where λ is the central wavelength of the bandgap and n is the diffraction order. That is, λ decreases when n increases. Therefore, the observed bandgap at near infrared and visible wavelength is the first order and second order, respectively. In addition, the experimental observation of multiple bandgaps in the reflection spectrum indicates the high optical quality for the fabricated DBR [Faraday Discuss., 2020, 223, 125, DOI: 10.1039/d0fd00026d].

The respective sentence in the manuscript has been corrected as below,

“As can be seen, when Sb₂S₃ is in an amorphous phase, DBR provides two PBGs, first order in the NIR wavelength (between 1100 and 1700 nm), and the second order in the visible wavelength (between 600 and 800 nm)” (Page 6)

It is well known that the bandwidth of the PBG decreases when the bandgap shifts to shorter wavelength [Sci Rep 13, 324 (2023), Faraday Discuss., 2020, 223, 125, DOI: 10.1039/d0fd00026d]. The frequency bandwidth (Δf_0) of a PBG can be calculated as, $\Delta f_0 = \left(\frac{4f_0}{\pi}\right) \sin^{-1}\left(\frac{n_2-n_1}{n_2+n_1}\right)$ with f_0 is the central frequency of the bandgap. It shows that Δf_0 increases with increasing f_0 . Therefore, the bandwidth (in wavelength) decreases with decreasing central wavelength of bandgap. In our case, the bandwidth of the NIR bandgap is almost double of that of the visible bandgap.

The following text has been included in the manuscript,

“The bandwidth of first order PBG is almost double of that of the second order. This is because the frequency bandwidth (Δf_0) of PBG is directly proportional to central frequency (f_0) of PBG, $\Delta f_0 = \left(\frac{4f_0}{\pi}\right) \sin^{-1}\left(\frac{n_2-n_1}{n_2+n_1}\right)$, where n_1 and n_2 is the refractive index of dielectric layers” (Page 6)

We have also corrected the nomenclature of TPP 1 as ‘second order TPP (TPP 2)’ and TPP 2 as ‘first order TPP (TPP 1)’ throughout the manuscript.

Comment 5: *The authors have studied the influence of incident angle and polarization of light (Fig. 1 and Fig. S2). They might discuss why they apply the normal incident rather than other angles in their sensing applications. The color in Fig. S2(a) seems not right*

Reply: We have studied the influence of incident angle and polarization of light to confirm the typical features of DBR and TPP cavities, which would be useful for other potential applications such as perfect absorbers, lasers, thermal emitters, etc. However, unpolarized light excitation at normal incidence is preferred for sensing, especially for point of care applications. The Raman

measurements are generally performed at normal incidence. In general, the oblique angle excitation and detection scheme is complex for microscope-based systems (Raman system).

Thank you for pointing out the color change in Fig. S2a. We have corrected it.

Comment 6: *In line 177, they claim that Fig. S4 shows the measured and calculated TPP modes. However, Fig. S4 only shows the calculated results.*

Reply: In the revised manuscript, this figure is Supplementary Fig. 5.

Comment 7: *In line 190, Fig. S5 should be replaced by Fig. S4*

Reply: We have corrected it.

Comment 8: *In line 234, the sample is annealed at 210°C to obtain the first TPP resonance at 785 nm wavelength. It is not consistent with Fig 3(b). In Fig. 3(b), the annealing temperature for 785 nm TPP is ~220°C*

Reply: As stated above (Reply to Comment 1), we have used 10 nm Au film coated TPP cavities for SERRS experiments. This sample provides TPP resonance around 785 nm by annealing at 210°C.

Comment 9: *What's the ratio of confined electric field between the TPP and reference cavities (Fig. S8 and Fig. S9)?*

Reply: We have calculated the confined electric field between the TPP and reference cavities at 785 nm by using 10 nm Au coated TPP cavities. The results are shown in Fig. R3.

Reviewer #2

I cannot recommend the publication of this manuscript in Nat. Comm. because the originality and novelty of the study reported are insufficient to justify the publication in it. This study is rather technical report not so much scientific. The idea of a tunable Tamm plasmon polariton (TPP) cavity based on a distributed Bragg reflector (DBR) on metal as a scalable sensing platform is interesting, but I do not think this technique will be used frequently in SERRS community. The authors reported “The development of a dynamically tunable plasmonic system would enable the precise matching of achieving SERRS by exactly matching the resonance wavelength with the absorption peak wavelength of biomolecules, thereby improving the sensitivity and the selectivity of SERRS.” However, usually in the case of SERRS the exact matching of the resonance wavelength with the absorption peak wavelength of a molecular is not always necessary. Moreover, the authors showed only one example of SERS (Figure 4) in the text, and they did not give enough interpretation. Therefore, I do not think they succeeded in demonstrate their method in measuring highly sensitive SERRS.

Reply: We would like to thank the reviewer for the time and effort reviewing our work. Below we provide a point-by-point response to the reviewer’s comments

Comment 1: *I cannot recommend the publication of this manuscript in Nat. Comm. because the originality and novelty of the study reported are insufficient to justify the publication in it. This study is rather technical report not so much scientific.*

Reply: We respectfully disagree with the reviewer’s comment. We provide a plethora of evidence below to clarify that our work is novel with scientific inputs.

1. Our work is the first demonstration of a tunable TPP cavity that shows very large TPP resonance wavelength tunability (295 nm) in the NIR wavelength. The tuning of TPP resonance was previously demonstrated by integrating TPP cavity with liquid crystal (LC) [*Physical Review Applied* **9**,064034 (2018)]. However, the system displayed very small TPP resonance tunability

(10 nm) because LC was integrated externally, and the cavity is very thick due to the LC layer [*Nanophotonics* **9**, 897–903 (2020)]. In particular, LC integrated cavities work on the principle of refractive index sensing, which means the TPP resonance shifts when the refractive index of the external LC layer changes with applied voltage and thus large resonance tunability is not possible. In a recent work, tunable TPP resonance in the mid infrared (MIR) spectral band was demonstrated by using cadmium oxide (CdO) as the plasmonic layer of the TPP cavity [*Nature Materials* **20** 1663-1669 (2021)]. This system also shows very small TPP resonance tunability by changing the carrier concentration of CdO. Also note that this system only works at MIR spectral band and optical pumping is required to tune the carrier concentration of CdO. **To realize a large resonance tunability, it is necessary to tune the photonic bandgap (PBG) of DBR since the TPP resonance is excited within the PBG. To realize this, we developed a novel PCM based tunable DBR and demonstrated large TPP resonance wavelength tunability by tuning the PBG of DBR.** Thus, we argue that our work is original with scientific inputs.

2. This is the first demonstration of SER(R)S using TPP concept: TPP cavities were previously proposed for different applications. Here we proposed a new application of TPP cavity for SERS. More importantly, the tunable and high resonance quality factor of the TPP mode can be utilized to realize multiplexed detection. The multiplexing assay is very important in sensing applications. To realize this, TPP resonance can be excited at commonly used Raman excitation wavelengths such as 633, 785 and 1064 nm and the resonance wavelength can be precisely tuned around these excitation wavelengths to match with absorption peak wavelength of different biomolecules (Fig. R1). In particular, the multiplexing properties of the presented sensor are based on the sensing capability of different TPP resonance wavelengths to corresponding absorption peak wavelength of biomolecules. Moreover, a higher dimensional biochemical analysis of complex samples with

multiple analytes is possible using multi-band SERS experiments under multiple excitation wavelengths, as analyte molecules have characteristic electronic transition features in addition to their vibrational fingerprints (*Adv. Funct. Mater.* 32, 2202231 (2022), *ACS Nano* 2, 2306 (2008)). However, the well-known metal nanoparticle (NP)-based SERS substrate cannot provide this feature because the NP usually shows broadband plasmonic resonance. On the other hand, the spectral tunability of TPP mode can also be used to mitigate the fabrication error.

3. The proposed scalable SERS substrate operates in the few-molecules detection regime. The development of ultrasensitive SER(R)S substrate highly depends on nanofabrication techniques (top-down and bottom-up), which limits its scalability for the practical realization of SER(R)S-based biosensors for real-time point-of-care applications. We showed that the proposed scalable TPP cavity, which is produced through thin film deposition without involving nano-structure patterning, is an ultrasensitive SERS substrate capable of detecting few biomolecules (<100 molecules).

Fig. R1 Calculated reflection spectrum of TPP cavity with tunable three TPP modes and (b) Measured reflection spectrum of TPP cavity (Amp) with two modes

The following text has been included in the discussion section of the revised manuscript:

“In addition, the excitation of multiple TPP modes at different excitation wavelengths can be utilized to realize multiplexed detection (see Supplementary Fig. 23), as such assay is essential in sensing applications because it can provide higher dimensional biochemical analysis of complex samples^{54, 55}”. (Page 18)

[54] Nie, M. *et al.* Broadband nanoscale surface enhanced Raman spectroscopy by multiresonant nanolaminate plasmonic nanocavities on vertical nanopillars. *Adv. Funct. Mater.* **32**, 2202231 (2022)

[55] Lutz, B. R. *et al.* Spectral analysis of multiplex Raman probe signatures. *ACS Nano* **2**, 2306 (2008)

Figure R1 has been included in the Supplementary information as Fig. 23

Comment 2: *The idea of a tunable Tamm plasmon polariton (TPP) cavity based on a distributed Bragg reflector (DBR) on metal as a scalable sensing platform is interesting, but I do not think this technique will be used frequently in SERRS community.*

Reply: As stated by the reviewer, the idea of the proposed scalable TPP cavity for SERRS application is interesting and its use varies depending on the sensing methods. It is well-known that the existing ultrasensitive SER(R)S substrates highly depend on top-down and bottom-up nanofabrication techniques, which limit their scalability and reproducibility. In this work, we propose a novel scalable and ultrasensitive SER(R)S substrate for commonly used longer wavelength Raman laser excitation (633, 785 and 1064 nm). More importantly, the high-quality factor TPP mode can be simultaneously excited at different laser wavelengths (Fig. R1) and the TPP resonance wavelength can be dynamically tuned. These narrowband TPP modes could be a potential candidate for multiplexed detection, however, it is a challenging task with existing SER(R)S substrates. In short, the proposed TPP cavity could be a promising scalable SERS substrate for ultrasensitive and multiplexed sensing applications.

Generally, label free SERS detection, where an analyte molecule is in direct contact with a substrate/nanoparticle and detected through the enhanced signal, is the most preferred sensing approach. But it is only feasible for molecules that possess strong Raman cross section, and, in our case, the detected protein is not Raman active and hence we used cy7 as reporter molecule and readout is achieved in a labelled manner. However, there is a great potential for developing SERS substrates with tunable plasmon spectrum as it opens new avenues for label free detection of those molecules that have innately moderate/low Raman cross section. In such a scenario, additional enhancement due to resonance effect is extremely useful to achieve the sensitivity.

By employing the tunability of the substrate, it is practical to detect a wide range of biomolecules in label free manner by just adopting the SERRS approach. This is practically relevant for those molecules possessing “moderate innate Raman cross section”.

The following text has been included in the revised manuscript,

“By employing the tunability of the substrate and the SERRS approach, it could be practically possible to detect a wide range of biomolecules in label free manner, particularly relevant for molecules that possess moderate innate Raman cross section.” (Page 18)

Comment 3: *The authors reported “The development of a dynamically tunable plasmonic system would enable the precise matching of achieving SERRS by exactly matching the resonance wavelength with the absorption peak wavelength of biomolecules, thereby improving the sensitivity and the selectivity of SERRS.” However, usually in the case of SERRS the exact matching of the resonance wavelength with the absorption peak wavelength of a molecule is not always necessary.*

Reply: We agree with the reviewer that the exact matching of the resonance wavelength with the absorption peak wavelength of a molecule is not always necessary to realize SERRS. However, it is suitable for quantitative and surface analysis [<https://doi.org/10.1016/B978-0-12-374413-5.00304-3>]. The similar effect will be more useful, when the plasmon spectrum of the substrate

matches with absorption of the dye and laser wavelength. It indicates that maximum SERRS, even up to single molecule detection is possible when these matching conditions are simultaneously met [<https://doi.org/10.1016/B978-0-12-374413-5.00304-3>].

As shown in Fig. 4a of the manuscript, reference sample cannot detect 1 μ M concentration of cy7.5 solution, where the laser excitation wavelength is only matched with cy7.5 absorption peak wavelength. However, TPP cavity is capable of detecting 10 nM concentration of cy7.5. It shows that the limit of detection increases by exactly matching the resonance wavelength with the absorption peak wavelength of the molecule. We have also estimated the adsorbed cTnI molecules on the illuminated sensor area for different concentrations of cTnI (see Supplementary Note 3), which shows that TPP cavity operates in the few-molecules detection regime.

To illustrate our point further, we highlight other works that demonstrated SERRS with high sensitivity by matching the resonance wavelength with the absorption peak wavelength of molecules as shown in Table R1, given below. (i) Rhodamine 6G molecules with attomolar levels of detection (10^{-18} M) has been reported by using silver nanoparticles and 514.5 nm laser excitation, which is approaching single molecule detection [<https://doi.org/10.1016/B978-0-12-374413-5.00304-3>], (ii) Zeptomole level detection of adenine molecules have been achieved using aluminum nanoparticle arrays and deep-UV laser ((257.2 nm) excitation [*J. Am. Chem. Soc.* **134**, 1966–1969 (2012)] and (iii) Highest enhancement factor for a 1 nm thick adenine monomer has been achieved using Al nanohole array and 266 nm laser excitation [*J. Am. Chem. Soc.* **143**, 19282–19286 (2021)]. In these works, ultrahigh sensitivity was achieved by combining SERRS with shorter wavelength laser excitation because enhancement factor is directly proportional to the fourth power of the excitation frequency. In general, nanopatterned and self-assembled

nanostructures were used in all the reported SERRS works. **It is important to note that our proposed TPP cavity achieved a femtomolar detection level (up to few molecules detection limit) using a near IR laser (785 nm) and scalable platform.**

Reference (Manuscript)	Platform	Scalable	Excitation wavelength (nm)	Plasmonic Resonance matching with absorption peak	Sensitivity
[17]	Al nanohole array	No	266	Yes	1 nm thick
[18]	Al nanoparticle array	No	257.2	Yes	Zeptomole
[19]	Ag nanoparticle	No	514.5	Yes	Attomolar
Our work	TPP cavity	Yes	785	Yes	Femtolar

Table R1 Comparison of TPP cavity with existing high sensitivity SERRS demonstrations

Also note that using our approach it is possible to clearly see all the prominent Raman peaks of cyanine dye molecules compared to previously reported resonant Raman effect, where only Raman excitation wavelength was matched with the absorption peak wavelength of dye [*Phys. Chem. Chem. Phys.* **9**, 6016–6020 (2007)]

The following text and reference have been included in the introduction section of the manuscript,

“Using this method, an enhancement factor of 10^8 -fold and above can be achieved by exciting at (i) the plasmonic resonance wavelength, (ii) the absorption maximum wavelength of biomolecules and (iii) the matching wavelength of plasmonic resonance and absorption maximum of biomolecules¹³⁻¹⁹. Among these, enhanced sensitivity with single molecule level detection has been realized by selecting the excitation frequency as the matching wavelength of plasmonic resonance and the absorption maximum of biomolecules¹⁷⁻¹⁹”. (Page 3)

[19] Smith, W. E. & Rodger, C. Surface Enhanced Raman Scattering (SERS), Applications, Encyclopedia of Spectroscopy and Spectrometry (Second Edition), (Academic Press 1999), Pages 2822-2827 (Page 25)

Comment 4: *Moreover, the authors showed only one example of SERS (Figure 4) in the text, and they did not give enough interpretation. Therefore, I do not think they succeeded in demonstrate their method in measuring highly sensitive SERRS*

Reply: As stated above, the exact matching of plasmonic resonance with absorption peak wavelength of the molecule improves the sensitivity and SERRS with ultrahigh sensitivity has been demonstrated using shorter wavelength laser sources. The motivation of our work is to demonstrate SERRS with enhanced sensitivity using longer wavelength lasers (633, 785 and 1064 nm). In Fig. R1, we showed the excitation of TPP mode at 633, 785 and 1064 nm simultaneously using Sb_2S_3 based cavity. For the proof-of-concept study, we selected 785 nm laser wavelength and performed two experiments (using cy7.5 dye and cy7 tagged cTnI proteins) to demonstrate the SERRS effect.

In the revised manuscript, we have estimated the number of adsorbed cTnI proteins on the illuminated sensor area and concluded that the proposed sensor operates in the few molecules detection regime. Since the aim of the proposed research is to demonstrate high sensitivity by matching the absorption peak wavelength of molecules with plasmonic resonance wavelength, we

believe that we have succeeded in demonstrating femtomolar detection level of protein using our scalable sensing platform. In the revised manuscript, we have included additional experimental results and calculations to support our claims such as,

- (i) Supplementary Note 3: Adsorbed molecule sensitivity,
- (ii) Supplementary Note 4: ELISA test for benchmarking
- (iii) Comparison of TPP cavity with other strategies (in supplementary information, Table 1 and Fig. 20)

Reviewer #3

The authors proposed a tunable Tamm plasmon polariton (TPP) cavity based on a distributed Bragg reflector (DBR) on metal as a scalable sensing platform for SERRS. The phase change materials used in the DBR allows for tuning of the operating wavelength. Furthermore, the authors demonstrated improved SERRS compared to reference measurements. Although the concept is novel and the demonstrations are solid, some major comments need to be addressed before further consideration.

Reply: We thank the reviewer for the high-level evaluation of our work, and for many insightful comments. Below we provide a point-by-point response to the reviewer's comments.

Comment 1: *Using the TPP mode to enhance the SERRS is novel. However, the performance of the TPP devices in terms of SERRS is not quantified and benchmarked properly. Unless the performance of TPP device is better or comparable to other strategies (demonstrations), I do not think the work meets the bar of nature comm*

Reply: Thank you for this important comment.

1. Quantification of adsorbed molecules

To quantify the adsorbed molecules sensitivity, we estimate the sensitivity of the Raman intensity shift to the number of cTnI molecules adsorbed on the sensor surface. Similar approach was previously used to quantify the adsorbed biomolecules on the sensor surface (*Nature Materials* 15, 621-627 (2016) and *Nature Communications* 9, 369 (2018)).

For each concentration c of the biomolecule in the sensor, there will be a maximum intensity shift $\Delta I(c)$ with respect to the bare sample. This shift is due to the presence of an average equilibrium population $N(c)$ of adsorbed cTnI molecules on the sensor surface, a number which cannot be directly measured. The sensitivity can be defined as $\Delta I(c)/N(c)$. Since $N(c)$ is unable to be

measured, here we estimate a reliable upper bound on this number, which is $N_{max}(c)$, a maximum number of biomolecules on average that can be adsorbed on the sensor surface. Note that the sensitivity, $\Delta I(c)/N_{max}(c)$ will be lower bound on the true sensitivity $\Delta I(c)/N(c)$ because $N(c) \leq N_{max}(c)$. First, we derive $N_{max}(c)$ by considering the illuminated beam area on the sensor surface. The illuminated beam diameter is around 2 μm , thus the effective sensor area is 4 μm^2 . It shows that only a small fraction of the total population of cTnI molecules will end up adsorbed on the illuminated sensor area. We will assume that the adsorbed molecules are equally distributed across the entire sensor surface, which has a dimension of 3 mm x 3 mm, an area of 9 mm^2 .

Hence, given a certain maximum possible adsorbed population on the surface, only a fraction $4 \mu\text{m}^2/9 \times 10^6 \mu\text{m}^2 = 0.444 \times 10^{-6}$ will be in the sensing area and relevant to the Raman intensity shift. Initially, there are $c (9 \text{ mm}^3 / 1\text{L}) \times 6.022 \times 10^{23} \text{ M}^{-1} = 5.42 c \times 10^{18} \text{ M}^{-1}$ biomolecules on the sensor. If all the molecules are to be adsorbed on the sensor surface, on average 0.444×10^{-6} of the total would be in the sensor area. Therefore, $N_{max}(c) = 24 c \times 10^{13} \text{ M}^{-1}$

In Fig. R1, we plotted N_{max} versus $\Delta I(c)$ for the measured values of c from 380 fM to 378 pM, corresponding to N_{max} ranging from 91 to 90720 molecules. The relationship between N_{max} and $\Delta I(c)$ is nonlinear, which is consistent with the following fitting function (red curve),

$$N_{max} = A_1(e^{\Delta I/I_1} - 1) + A_2(e^{\Delta I/I_2} - 1)$$

where $A_i, I_i, i=1, 2$, are the fitting parameters. The best fitting values are $A_1=2, A_2=9 \times 10^2, I_1=350$, and $I_2=900$. Since there are clearly two exponential regimes in the data, a biexponential fitting function is required. The fitting function is not just a sum of two exponentials but includes a constant term - (A_1+A_2) so that $N_{max}=0$ when $\Delta I = 0$. The observed nonlinearity is due to the

possibility of multiple adsorbed molecules on the sensor leads to interference effects (when c increases) and hence, with each additional molecule having a decreasing impact on the Raman intensity shift. The obtained sensitivity ($\Delta I(c)/N_{max}(c)$) is 0.692 for 380 fM and this value decreases with increasing c due to nonlinearity. According to this analysis, it can be concluded that the sensor operates in the few molecules (<100) detection regime.

Fig. R1 The maximum number of cTnI molecules adsorbed in the illuminated sensor area versus the corresponding Raman intensity shift with respect to bare sample

This section has been included in the Supplementary information as Supplementary Note 3: Adsorbed molecule sensitivity

The following text and references have been included in the manuscript to address this point:

“The sensitivity of the Raman intensity shift ($\Delta I(c)$) to the number of cTnI molecules (N_{max}) adsorbed on the illuminated sensor area is estimated using the method described in Ref [47, 48] (see Supplementary Note 3). In Fig. 4e, we plot N_{max} versus $\Delta I(c)$ for the measured values of concentration (c) from 380 fM to 378 pM, corresponding to N_{max} ranging from 91 to 90720 molecules. The relationship between N_{max} and $\Delta I(c)$ is nonlinear, which is consistent with a fitting

function (red curve). The observed nonlinearity is due to the possibility of multiple adsorbed molecules on the sensor leads to interference effects (when c increases) and hence, with each additional molecule having a decreasing impact on the Raman intensity shift” (Page 16)

[47] Sreekanth, K. V. *et al.* Extreme sensitivity biosensing platform based on hyperbolic metamaterials. *Nat. Mater.* **15**, 621-627 (2016)

[48] Sreekanth, K. V. *et al.* Biosensing with the singular phase of an ultrathin metal-dielectric nanophotonic cavity. *Nat. Commun.* **9**, 369 (2018)

Figure R1 has been included in the revised manuscript as Fig. 4e

2. ELISA test for benchmarking

We have conducted an ELISA test for benchmarking the performance of the proposed TPP cavity-based SERS substrate. In Table R1, Column 1 represents the actual known concentration of cTnI protein spiked in plasma. Column 2 is its corresponding measured ELISA absorbance. Final concentration of cTnI protein (shown in column 3) detected by ELISA after interpolating from the calibration plot (shown in Fig. R2). Due to the limitation with ELISA kit, we could detect in the range of ~ 166 pM to 2.6 pM. Accordingly, higher concentrations of spiked cTnI samples were prepared and studied followed by calibration and normalization to obtain the calculated spiked cTnI concentration detected by ELISA. The ELISA results were comparable for higher concentration (pM), but ELISA kit was not able to provide any results for lower concentrations (fM).

Figure R2 shows the calibration plot of ELISA with interpolated spiked cTnI samples (in red). As can be seen, the sensitivity of the TPP-SERS platform for the protein detection is comparable with the gold standard, ELISA method for higher concentration. Note that ELISA could only detect up to 2.6 pM, which is much higher cTnI protein concentration than what was detected using TPP

substrate, which is in fM range. In addition, ELISA measurement takes much longer time and involves multiple washing steps. SERS methods provide a relatively faster means for sensitive detection.

Actual cTnI concentration	Absorbance	cTnI detected by ELISA
378 pM	1.9525	330 pM
38 pM	1.1272	41 pM
380 fM	0.498	0
190 fM	0.205	0
95 fM	0.403	0
47.5 fM	0.26	0

Table R1 Detection of spiked cTnI by ELISA

Fig. R2 Calibration plot of ELISA with interpolated spiked cTnI samples (in red)

This section has been included in the Supplementary information as Supplementary Note 4: ELISA test for benchmarking

The following text has been included in the manuscript to address this point:

“We detected spiked cTnI by ELISA test and found that the sensitivity of the TPP-SERS platform for the protein detection is comparable with that of the gold standard (see Supplementary Note 4)”
(Page 16)

3. Comparison of TPP cavity with other strategies

To compare the performance of TPP cavity with other strategies (demonstrations), we discuss the key features of existing best SERRS demonstrations and our TPP cavity in Table R2.

Reference (Manuscript)	Platform	Scalable	Excitation wavelength (nm)	Plasmonic Resonance matching with absorption peak	Sensitivity
[17]	Al nanohole array	No	266	Yes	1 nm thick
[18]	Al nanoparticle array	No	257.2	Yes	Zeptomole
[19]	Ag nanoparticle	No	514.5	Yes	Attomolar
Our work	TPP cavity	Yes	785	Yes	Femtomolar

Table R2 Comparison of TPP cavity with existing SERRS demonstrations

SERRS can be realized by exciting at (i) the plasmonic resonance wavelength, (ii) the absorption maximum wavelength of biomolecules and (iii) the matching wavelength of plasmonic resonance and absorption maximum of biomolecules [19]. Among these, enhanced sensitivity with single

molecule level detection is possible by selecting the excitation frequency as the matching wavelength of plasmonic resonance and the absorption maximum of biomolecules [17-19]. In [17], the highest enhancement factor for a 1 nm thick adenine monomer has been achieved using Al nanohole array and 266 nm laser excitation. Zeptomole level detection of adenine molecules have been achieved using aluminum nanoparticle arrays and deep-UV laser (257.2 nm) excitation [18]. Rhodamine 6G molecules with attomolar levels of detection (10^{-18} M) have been reported by using silver nanoparticles and 514.5 nm laser excitation, which is approaching single molecule level detection [19]. In all works, ultrahigh sensitivity was achieved by combining SERRS with shorter wavelength laser excitation because enhancement factor is directly proportional to the fourth power of the excitation frequency. In addition, nanopatterned and self-assembled nanostructures were used in all the reported SERRS works (Table R2). **It is important to note that our proposed TPP cavity achieved a femtomolar detection level (few molecules detection limit) using a near IR laser (785 nm) and a scalable platform. Also note that the sensitivity > femtomolar can be achievable by nanopatterning the top Au layer and even using a 785 nm laser, which is comparable with existing other strategies [17-19].** More importantly, the excitation of high resonance quality factor multiple TPP modes with continuous tuning of each mode is possible by changing the phase of Sb₂S₃ layers, which could be useful of multiplexed detection (by commonly using laser wavelengths of Raman system (633, 785 and 1064 nm), see Reply to comment 3.

In addition, we have conducted SERS experiments by anchoring cy7 tagged-cTnI onto 60 nm Au nanoparticles (here, we used 38 pM concentrations of cTnI). As shown in Fig. R3, the sensitivity is poor as there is no SERRS effect with respect to plasmon absorption of nanoparticles (~532 nm), which is far away from laser excitation wavelength at 785 nm and absorption spectrum of dye.

The following text has been included in the manuscript to address this point:

“We found that the sensitivity of TPP cavity is much higher compared to signal achieved using 60 nm Au colloids (see Supplementary Fig. 20). It can also be noted that the performance of the TPP cavity is comparable with that of reported metallic nanostructures based SERS demonstrations (see Supplementary Table 1)” (Page 16)

Table R2 and Fig. R3 has been included in the supplementary information as Table 1 and Figure 20, respectively.

Fig. R3 SERS spectra of cy7 tagged cTnI protein (38 pM) in Au colloid. A SEM image of ~60 nm diameter Au nanoparticles is shown (inset).

Comment 2: Based on my understanding of Phase change materials (PCMs), it is extremely challenging (if not impossible) to amorphized large PCM areas (~30 um square is the largest size I'm aware of, Abdollahramezani, Sajjad, et al. "Electrically driven reprogrammable phase-change metasurface reaching 80% efficiency." Nature communications 13.1 (2022): 1696.). Thus, the authors need to be careful about the wording in the discussion sections regarding the amorphization process.

Reply: We agree with reviewer that electrical reversible switching of large area PCM based devices is extremely challenging. Since our proposed TPP cavity is a multilayer film with thickness $\approx 1.5\mu\text{m}$ and consisting of many alternating PCM layers, it is a challenging task to perform reamorphization process using electrical pulses. As suggested, the respective text in the manuscript has been amended as below,

“Since electrically tunable devices are more feasible for practical applications of tunable TPP cavities, we reported our initial results on electrically continuous forward tuning of TPP resonance over a uniform switching area of $25\mu\text{m} \times 100\mu\text{m}$ using a microheater-integrated TPP cavity. More importantly, electrical reversible switching of Sb_2S_3 based cavities is necessary for reconfigurable photonic device applications, however, the efficient switching of large areas of thick ($>1\mu\text{m}$) PCM-based devices is a challenging task⁵¹⁻⁵³”. (Page 17)

Comment 3: *Following the previous comment, in my opinion, adding the PCM is a great idea, allowing the device to be tuned or adjusted towards longer wavelength, so that fabrication error can be mitigated. The authors mentioned that the device used for SERRS was annealed at 210 C, is it to shift the resonance to compensate for fabrication error? I recommend the authors highlight this aspect*

Reply: Thank you for this important comment. As suggested by the reviewer, spectral tunability of TPP mode can also be used to mitigate the fabrication error. Since we used only one Raman excitation wavelength (785 nm), this point is valid, as the principle of SERRS is to exactly match the resonance wavelength with the absorption peak wavelength of biomolecules. However, in a broader sense, the tunable spectral range of TPP mode can be used to detect different biomolecules based on SERRS (by using different excitation wavelengths) and also for the detection of a wide range of biomolecules in label free manner. As shown in Fig. R4, the excitation of multiple TPP modes with continuous tuning of each mode is possible by changing the phase of Sb_2S_3 layers,

which could be useful of multiplexed detection (by commonly using laser wavelengths of Raman system (633, 785 and 1064 nm).

Moreover, a higher dimensional biochemical analysis of complex samples with multiple analytes is possible using multi-band SERS experiments under multiple excitation wavelengths, as analyte molecules have characteristic electronic transition features in addition to their vibrational fingerprints (*Adv. Funct. Mater.* 32, 2202231 (2022), *ACS Nano* 2, 2306 (2008)). However, the well-known metal nanoparticle (NP)-based SERS substrate cannot provide this feature because the NP usually shows broadband plasmonic resonance.

Generally, label free SERS detection, where an analyte molecule is in direct contact with a substrate/nanoparticle and detected through the enhanced signal, is the most preferred sensing approach. But it is only feasible for molecules that possess strong Raman cross section, and, in our case, the detected protein is not Raman active and hence we used cy7 as reporter molecule and readout is achieved in a labelled manner. However, there is a great potential for developing SERS substrates with tunable plasmon spectrum as it opens new avenues for label free detection of those molecules that have innately moderate/low Raman cross section. In such a scenario, additional enhancement (up to 2 orders provided by resonance effect) is extremely useful to achieve the sensitivity.

By employing the tunability of the substrate, it is practical to detect a wide range of biomolecules in label free manner by just adopting the SERRS approach. This is practically relevant for those molecules possessing “moderate innate Raman cross section”.

Fig. R4 Calculated reflection spectrum of TPP cavity with tunable three TPP modes and (b) Measured reflection spectrum of TPP cavity (Amp) with two modes

The following text has been included in the manuscript to address the reviewer suggestion,

“By employing the tunability of the substrate and the SERRS approach, it could be practically possible to detect a wide range of biomolecules in label free manner, particularly relevant for molecules that possess moderate innate Raman cross section.” (Page 18)

“The spectral tunability of TPP mode can also be used to mitigate the fabrication error.” (Page 18)

“In addition, the excitation of multiple TPP modes at different excitation wavelengths can be utilized to realize multiplexed detection (see Supplementary Fig. 23), as such assay is essential in sensing applications because it can provide higher dimensional biochemical analysis of complex samples^{54, 55}.” (Page 18)

[54] Nie, M. *et al.* Broadband nanoscale surface enhanced Raman spectroscopy by multiresonant nanolaminate plasmonic nanocavities on vertical nanopillars. *Adv. Funct. Mater.* **32**, 2202231 (2022)

[55] Lutz, B. R. *et al.* Spectral analysis of multiplex Raman probe signatures. *ACS Nano* **2**, 2306 (2008)

Figure R4 has been included in the Supplementary information as Fig. 23

Comment 4: *As the authors suggested, the electric heating is not uniform. Please include thermal images or simulations to more quantitatively discuss: (1) how non-uniform the temperature is; (2) how does the non-uniform temperature affect the measurement (the sampling domain is only 100 by 100 μm , does it only affect the reproducibility? Or it also makes the collected region non-uniform?)*

Reply: Thank you for the comment. As stated in the Methods section of the manuscript, “The normal incidence reflectance measurements were performed using a microspectrophotometer (Jasco, MSV-5200) with a sampling domain size of $100\ \mu\text{m}\times 100\ \mu\text{m}$ and $25\ \mu\text{m}\times 25\ \mu\text{m}$ for scalable and nanopatterned samples, respectively”. We would like to clarify that sampling domain size of $25\ \mu\text{m}\times 25\ \mu\text{m}$ was also used to measure the reflection spectra of the electrically tunable cavity (Fig.3d) because the linewidth of the microheater bar is $25\ \mu\text{m}$ (Fig. R5). However, all other normal incidence reflection measurements were performed using a beam area of $100\ \mu\text{m}\times 100\ \mu\text{m}$, where the samples were annealed using a hot plate so that large area forward switching (amorphous to crystalline) was realized. We have corrected the respective text in the Methods section.

Thank you for the suggestion of acquiring thermal images or performing simulations to further understand the non-uniform temperature distribution on the cavity. However, thermal images or thermal simulations is tricky for the proposed TPP cavity since it is a thick multilayer film consisting of many alternating PCM and SiO_2 layers. Here we discuss the non-uniform temperature distribution in a more efficient manner by monitoring the color change of Sb_2S_3 - SiO_2 DBR sample with applied DC current. For this purpose, initially tungsten ($200\ \text{nm}$) microheater was fabricated on Si substrate and followed by the deposition of 10 alternating layers of Sb_2S_3 ($170\ \text{nm}$) and SiO_2 ($100\ \text{nm}$) on the entire area of the Si substrate. Figure R5 shows the optical microscopic image of ‘as-deposited’ ($0\ \text{mA}$) and electrically annealed ($320\ \text{mA}$) Sb_2S_3 - SiO_2 DBR. It is clear that color change is not uniform throughout the microheater bar due to non-

uniform temperature distribution, however, uniform color change is possible for a $25\ \mu\text{m} \times 100\ \mu\text{m}$ sample area. As the reviewer stated in Comment 2 and according to data presented in Fig. R5, we confirm that uniform temperature distribution is obtained for a measurement area of $25\ \mu\text{m} \times 25\ \mu\text{m}$. Also note that in our recent work the electrical switching of $500\ \mu\text{m} \times 500\ \mu\text{m}$ pixel size was demonstrated (*Nano Letters* 23, 5236 (2023)). Thus, the switching area can be further widened by using proper microheater designs.

Fig. R5 Optical microscope image showing the tunable color of $\text{Sb}_2\text{S}_3\text{-SiO}_2$ DBR with applied DC current

The following text has been included in the manuscript to address this point,

“Since the linewidth of the microheater bar is $25\ \mu\text{m}$, a sample domain size of $25\ \mu\text{m} \times 25\ \mu\text{m}$ was used for the reflectance measurement” (Page 12)

“We also confirmed that the temperature distribution is uniform over a sample area of $25\ \mu\text{m} \times 100\ \mu\text{m}$ by monitoring the color change of DBR with applied current (see Supplementary Note 1 and Supplementary Fig. 8)” (Page 12)

Figure R5 has been included in the Supplementary information as Fig. 8

The following section has been added in the Supplementary information as **Supplementary Note 1: Uniform temperature distribution on the switching area**

“Here we demonstrate the uniform temperature distribution on the measured sample area by monitoring the color change of Sb_2S_3 - SiO_2 DBR with applied DC current. For this purpose, initially tungsten (200 nm) microheater was fabricated on Si substrate and followed by the deposition of 10 alternating layers of Sb_2S_3 (170 nm) and SiO_2 (100 nm) on the entire area of the Si substrate. Supplementary Figure 8 shows the optical microscopic image of ‘as-deposited’ (0 mA) and electrically annealed (320 mA) Sb_2S_3 - SiO_2 DBR. It is clear that the color change is not uniform throughout the microheater bar due to non-uniform temperature distribution, however, uniform color change is possible for a sample area of $25 \mu\text{m} \times 100 \mu\text{m}$. The switching area can be further widened by using proper microheater designs”.

Following reference has been added in the manuscript as Ref [41]

[41] Prabhathan, P. et al. Electrically tunable steganographic nano-optical coatings. *Nano Lett.* **23**, 5236–5241(2023)

Comment 5: *The loss of Sb_2S_3 is quite high at $\sim 785 \text{ nm}$. In Fig. 1 b and c, the first DBR reflection band is almost smashed after switching to crystalline state. Please comment on the loss from Sb_2S_3 , how it changes the SERRS performance.*

Reply: As clearly pointed out by the reviewer, the first DBR reflection band is almost smashed after switching to crystalline state, where the sample was annealed at 250°C . This is because the loss of Sb_2S_3 layers increases with crystallization. To investigate the loss of Sb_2S_3 at different annealing temperatures, we measured the extinction coefficient (k) of 170 nm thick Sb_2S_3 film at temperature of 200°C , 210°C and 220°C and plotted along with the k values of amorphous and crystalline films, as shown in Fig. R6a. It is evident that the obtained k value at 785 nm is negligible for the samples annealed at $\leq 220^\circ\text{C}$.

Fig. R6 (a) Measured extinction coefficient of Sb_2S_3 thin film at different temperatures. (b) Measured reflection spectrum of 10 nm Au coated TPP cavity (as deposited and annealed at 210°C)

In fact, the loss of the Sb_2S_3 layer affects the SERRS performance. The resonance quality factor of the TPP mode decreases with increasing loss. However, we used 10 nm Au film for SERRS studies. This sample initially shows the TPP resonance at 760 nm for ‘as deposited’ sample and then the resonance at 785 nm is obtained by annealing at 210°C (Fig. R1b). It is evident that the experimentally obtained resonance quality factor of the 10 nm Au coated TPP cavity (annealed at 210° C) is slightly decreased compared to amorphous cavity. This is due to the increased surface roughness of the TPP cavity by annealing even though the loss of Sb_2S_3 is zero at 785 nm (Fig. R6b). Also note that the surface roughness increases with decreasing the Au thickness. This is the reason why we used an optimized thickness of 10 nm Au layer to obtain high surface roughness as well as a considerable resonance quality factor at 785 nm.

We have also performed FDTD simulations to estimate the enhancement of electric field at 785 nm by using 10 nm thick Au film and the optical constants of Sb_2S_3 film that annealed at 210°C (see Supplementary Fig. 13). In order to utilize the full tunable spectral range of TPP cavity for

SERRS, it is recommended to select a Raman excitation wavelength where the loss of Sb_2S_3 is negligible in both amorphous and crystalline phases, for example 1064 nm (Fig. R7).

Fig. R7 Excitation of high-quality factor TPP mode in both amorphous and crystalline phases

Figure R6a and Fig. R6b has been included in the supplementary information as Fig. 1b and Fig. 9, respectively.

The following text has been included in the manuscript,

“Also note that the loss in Sb_2S_3 slightly increases with increase in annealing temperature from the amorphous phase (see Supplementary Fig. 1b)” (Page 6)

“Using this approach, the surface roughness of the TPP cavity is greatly enhanced (see Supplementary Fig. 10). As a result, the minimum intensity at the TPP resonance wavelength (785 nm) is slightly increased” (Page 13)

Minor Comments

Comment 1: Line 61, the authors stated that 10^6 enhancement by SERRS has been demonstrated, but the authors did not quantify the enhancement by the TPP structure

Reply: Thank you for the comment. The enhancement factor of TPP substrate has been quantified using the well-known methodology. In our experiment, we compared 100 μM cyanine 7.5 (cy7.5) based SERS intensity with a thin liquid layer of cy7.5. The spatially averaged enhancement factor, G , is determined using a well-established method;

$$G = (H * \rho / (R * \mu)) * (I_{\text{SERS}} / I_{\text{RAMAN}}) \quad (1)$$

The estimated enhancement factor of the TPP substrate is $\sim 3.52 \times 10^7$.

The following text and references have been included in the revised manuscript,

“The enhancement factor of TPP substrate is quantified using the well-known methodology^{49, 50} and the estimated enhancement factor is $\sim 3.52 \times 10^7$ ” (Page 16)

[49] Dinish, U. S., Yaw, F. C., Agarwal, A. & Olivo, M. Development of highly reproducible nanogap SERS substrates: comparative performance analysis and its application for glucose sensing. *Biosens. Bioelectron.* **26**, 1987 (2011)

[50] Perumal J. *et al.* Design and fabrication of random silver films as substrate for SERS based nano-stress sensing of proteins. *RSC Adv.* **4**, 12995 (2014)

Comment 2: *Line 163, it is not immediately clear to readers why the sensing application requires the metal side contacted with the external environment. I recommend mentioning the field-profiles here*

Reply: As stated in the manuscript, two scalable configurations can be used for the excitation of TPP (i) a thin metal layer on top of the DBR (air-metal-DBR) and (ii) a thin metal layer between DBR and substrate (DBR-metal-substrate). In TPP cavities, field intensity tightly confines at DBR/metal interface and decays exponentially to metal as well as to the DBR. In order to utilize this decaying field for sensing (refractive index and SERS), the biomolecules should be in direct contact with the metal. As shown in Fig. R3a, the intensity is tightly confined at $\text{Sb}_2\text{S}_3/\text{Au}$ interface

and decays to the Au layer for TPP cavity. Moreover, metal such as Au is biocompatible so that biomolecules can easily be functionalized on the sensor surface using well-known surface chemistry approaches. Thus, air-metal-DBR configuration is preferred for sensing applications. As suggested, the respective text in the manuscript has been amended as below,

“For sensing applications, air-metal-DBR configuration is more preferred because the analytes can directly experience the decaying near field on the metal layer” (Page 8)

REVIEWER COMMENTS

Reviewer #1 (Remarks to the Author):

The authors answer my questions about this manuscript in detail.

Reviewer #3 (Remarks to the Author):

The authors provided a very comprehensive response letter, and most of my concerns were addressed. I can recommend the publication in Nature Comm after the two questions are addressed:

(1) Please also provide surface roughness data of the sample before annealing.

(2) The enhancements are typically correlated with $|E|^4/|E_0|^4$. The value is estimated to be $\sim 2-5^4$, which is 256-625 in this case. However, the observed Raman enhancement is $\sim 10^7$. Please provide more explanations on the observed Raman enhancements.

a. Reference:

i. Sprague-Klein, Emily A., et al. "Photoinduced plasmon-driven chemistry in trans-1, 2-bis (4-pyridyl) ethylene gold nanosphere oligomers." *Journal of the American Chemical Society* 140.33 (2018): 10583-10592.

ii. Langer, Judith, et al. "Present and future of surface-enhanced Raman scattering." *ACS nano* 14.1 (2019): 28-117.

POINT-BY-POINT RESPONSE

Reviewer #1

The authors answer my questions about this manuscript in detail.

Reply: We thank the reviewer for favorable evaluation of our work.

Reviewer #2

The authors provided a very comprehensive response letter, and most of my concerns were addressed. I can recommend the publication in Nature Comm after the two questions are addressed

Reply: We thank the reviewer for the high-level evaluation of our work, and for extra insightful comments. Here, we have provided detailed answers to the comments and used the feedback to suitably revise the manuscript accordingly.

Comment 1: *Please also provide surface roughness data of the sample before annealing*

Reply: Thank you for the comment. The SEM image of the topology of the TPP cavity before annealing is shown in Fig. R1a. We also put the SEM image of the topology of the TPP cavity after annealing shown in Supplementary Figure 10c at the same magnification as Fig. R1b for

easier comparison. We measured the surface roughness (R_a) value of the 10 nm Au coated TPP cavity using AFM, which is 4.48 nm before annealing and 8.4 nm after annealing. It is evident that the surface roughness of the TPP cavity is increased after annealing at 210°C.

Fig. R1. SEM image of the topology of the TPP cavity (a) before annealing and (b) after annealing

To address this point, Fig. R1a is added to Supplementary Figure 10 and the figure caption has been amended as,

“Supplementary Figure 10 | SEM images of the topology of the TPP cavity for different magnifications (a) before annealing and (b, c & d) after annealing. (e) AFM image of annealed TPP sample (measured surface roughness, $R_a = 8.4$ nm)” (Page 6 in Supplementary Information)

Comment 2: *The enhancements are typically correlated with $|E|^4/|E_0|^4$. The value is estimated to be $\sim 2 \cdot 5^4$, which is 256-625 in this case. However, the observed Raman enhancement is $\sim 10^7$. Please provide more explanations on the observed Raman enhancements.*
Reference: *i. Sprague-Klein, Emily A., et al. "Photoinduced plasmon-driven chemistry in trans-1,2-bis(4-pyridyl) ethylene gold nanosphere oligomers." Journal of the American Chemical Society*

140.33 (2018): 10583-10592.ii. Langer, Judith, et al. "Present and future of surface-enhanced Raman scattering." *ACS nano* 14.1 (2019): 28-117

Reply: Thank you for this important comment. As clearly pointed out by the reviewer, the enhancement factor (EF) is typically correlated with $|E|^4/|E_0|^4$. **However, this method is usually used to calculate the EF from a single localized hot spot** (*Ru et al., J. Phys. Chem. C* 111, 13794–13803, 2007). Also note that cross-sectional electric field intensity distribution ($I=|E|^2$) of the entire TPP cavity is shown in Supplementary Figure 22. Therefore, the intensity values acquired from this figure cannot provide the exact EF of the TPP cavity.

Since the proposed TPP cavity surface consists of random nanostructures, we calculate the spatially averaged EF of the sample using a well-established method [1-4]. In our experiment, we compared 100 μM cyanine 7.5 (cy7.5) based SERS intensity with a thin liquid layer of cy7.5.

$$EF = (I_{\text{SERS}}/N_{\text{SERS}})/(I_{\text{Bulk}}/N_{\text{Bulk}}) \quad (1)$$

In Eq. (1), I_{bulk} and I_{SERS} represent the intensity values at the scattering band of interest (e.g., 943 cm^{-1}) in the bulk liquid Raman spectrum and SERS spectrum, respectively. N_{Bulk} corresponds to the number of cy7.5 molecules in the bulk solution contributing to the unenhanced Raman signal, while N_{SERS} refers to the number of chemisorbed cy7.5 molecules on the substrate contributing to the SERS signal.

The values of N_{SERS} and N_{Bulk} can be determined using the following equations:

$$N_{\text{SERS}} = A_{\text{beam}} * R * \mu \quad (2)$$

$$N_{\text{Bulk}} = A_{\text{beam}} * H * \rho \quad (3)$$

where A_{beam} represents the area of the laser beam, R is the fractional surface area (ratio of exposed surface area to the nanostructure unit cell), μ is the packing density of cy7.5 molecules on the surface of the substrate, H is the apparent height of the cy7.5 liquid layer emitting the Raman signal, and ρ is the molecular density of the prepared cy7.5 solution.

EF can be written as:

$$EF = (H * \rho / (R * \mu)) * (I_{\text{SERS}} / I_{\text{Bulk}}) \quad (4)$$

In our case, the estimated values of parameters used in Eq. (4) are $\mu = 6.261 \times 10^{18}$ molecules/cm², $R = 0.325$, $H = 16 \times 10^{-4}$ cm and $\rho = 6.023 \times 10^{21}$ molecules/cm³, I_{SERS} (normalized to laser power and concentration) = 16779×10^5 counts, $I_{\text{Bulk}} = 226$ counts. The estimated enhancement factor of the TPP substrate is $\sim 3.52 \times 10^7$.

References

1. Cai, W. B., *et al.* Investigation of surface-enhanced Raman scattering from platinum electrodes using a confocal Raman microscope: dependence of surface roughening pretreatment *Surf Sci.* 406, 9-22 (1998)
2. Smythe, E. J., Dickey, M. D., Bao, J., Whitesides, G. M., Capasso, F. Optical Antenna Arrays on a Fiber Facet for in Situ Surface-Enhanced Raman Scattering Detection *Nano Lett.*, 9 1132-1138 (2009)
3. Dinish, U. S., Yaw, F. C., Agarwal, A. & Olivo, M. Development of highly reproducible nanogap SERS substrates: comparative performance analysis and its application for glucose sensing. *Biosens. Bioelectron.* 26, 1987 (2011)
4. Perumal J., *et al.* Design and fabrication of random silver films as substrate for SERS based nano-stress sensing of proteins. *RSC Adv.* 4, 12995 (2014)

This section has been included in Supplementary Information as Supplementary Note 4: Enhancement Factor calculations. (Page 16)

REVIEWERS' COMMENTS

Reviewer #3 (Remarks to the Author):

The authors have addressed my comments and I recommend the publication.

Reviewer #2

The authors have addressed my comments and I recommend the publication.

Reply: We thank the reviewer for favorable evaluation of our work.